# Experiences and Challenges in Fatality Reduction on Polish Roads

**Kazimierz Jamroz** 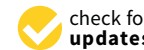**, Marcin Budzyński** , **Aleksandra Romanowska** * , **Joanna Żukowska** , **Jacek Oskarbski** and **Wojciech Kustra**

Faculty of Civil and Environmental Engineering, Gdansk University of Technology, 80-233 Gdańsk, Poland; kjamroz@pg.edu.pl (K.J.); mbudz@pg.edu.pl (M.B.); joanna@pg.edu.pl (J.Ż.); joskar@pg.edu.pl (J.O.); castrol@pg.edu.pl (W.K.)

* Correspondence: aleksandra.romanowska@pg.edu.pl; Tel.: +48-58-347-17-97

**Abstract:** According to the UN, road safety is the key to achieving sustainable development goals, yet the complexity of how road accidents happen makes this a difficult challenge leaving many countries struggling with the problem. For years, Poland has infamously been one of the EU's top countries for road-accident fatality rates. Despite that, it has made significant progress in the last thirty years with a fatality reduction of more than 60%. A number of factors have contributed to this result: improving the socioeconomic situation, improving road safety measures, changing road user behaviour and changing national road safety programmes. This article presents Poland's approach to road safety and, in particular, Vision Zero, adopted in 2005. Poland's road safety changed over the years as the country learned from its successes and failures. Tools for forecasting fatalities were developed and used to identify the main factors that have helped to reduce deaths. An assessment was conducted on how Poland could implement Vision Zero until 2050 under different road safety scenarios. It was found that in order to achieve the EU's goal for 2030, Poland must reduce fatalities to 1200. While it is an ambitious goal, it is also an important step towards zero fatalities in 2050.

**Keywords:** road safety; vision zero; sustainable transport safety; sustainable development goals

## 1. Introduction

Improving public health is an essential goal for the world's sustainable development as indicated by the United Nations (UN) [1]. A major public health issue for society today, traffic injuries are caused by malfunctioning road transport systems. The problem is global and exists in both developing and industrialised countries. International road safety comparisons can be made using the demographic road fatality rate (RFR). Based on this indicator and the literature [2–9], it can be assumed that a country's road safety level changes non-linearly and follows its socioeconomic development measured with the Gross Domestic Product per capita (GDPPC) (Figure 1). In the initial phase of socioeconomic development, the RFR increases until it reaches the breakpoint, which occurs for low GDPPC rates and when RFR is the highest. After the breakpoint, RFR decreases while GDPPC continues to grow. In this context, developing countries are in the upward phase of the indicator (before the breakpoint) and industrialised countries are in the downward phase (having gone past the breakpoint). With Poland in the second group, it is critical to maintain the downward fatality trend and possibly accelerate the rate of reduction.

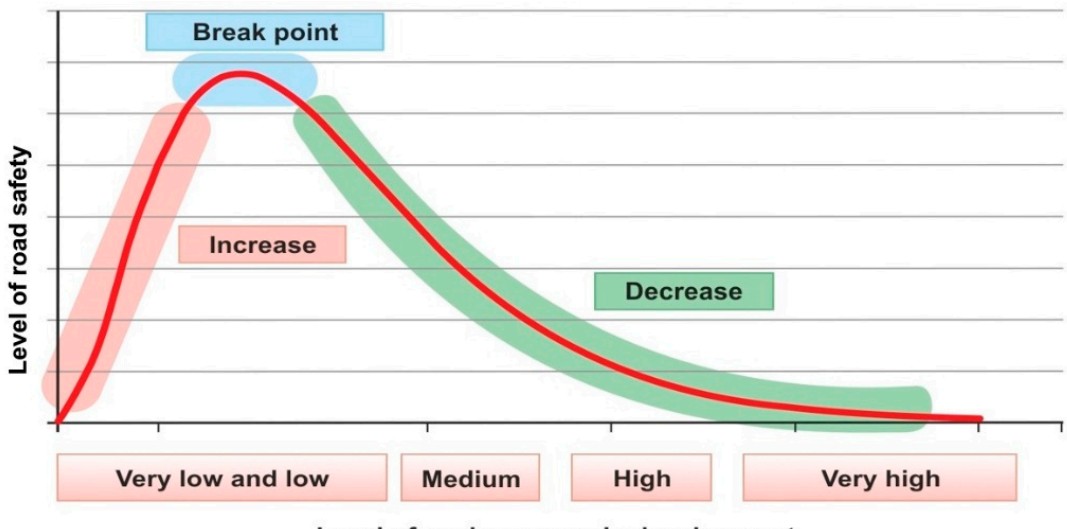

**Figure 1.** A model describing the relation between road safety and the level of a country's socioeconomic development.

Since 1991 (the breakpoint of the upward trend), Poland's road safety has improved significantly. Fatalities are down from the catastrophic level of nearly 8000 road deaths in 1991 to less than 3000 in 2017 (Figure 2). The reduction could be achieved thanks to an improving socioeconomic situation after the political change, road safety efforts, the changing behaviour of road users and the implementation of national road safety programmes. A major breakthrough in the approach to road safety came in 2005 which saw the launch of GAMBIT 2005, Poland's National Road Safety Programme for the years 2005–2007–2013. The programme adapted Sweden's Vision Zero as an ethically justified vision of road safety [10], marking the beginning of Poland's systemic approach to road safety.

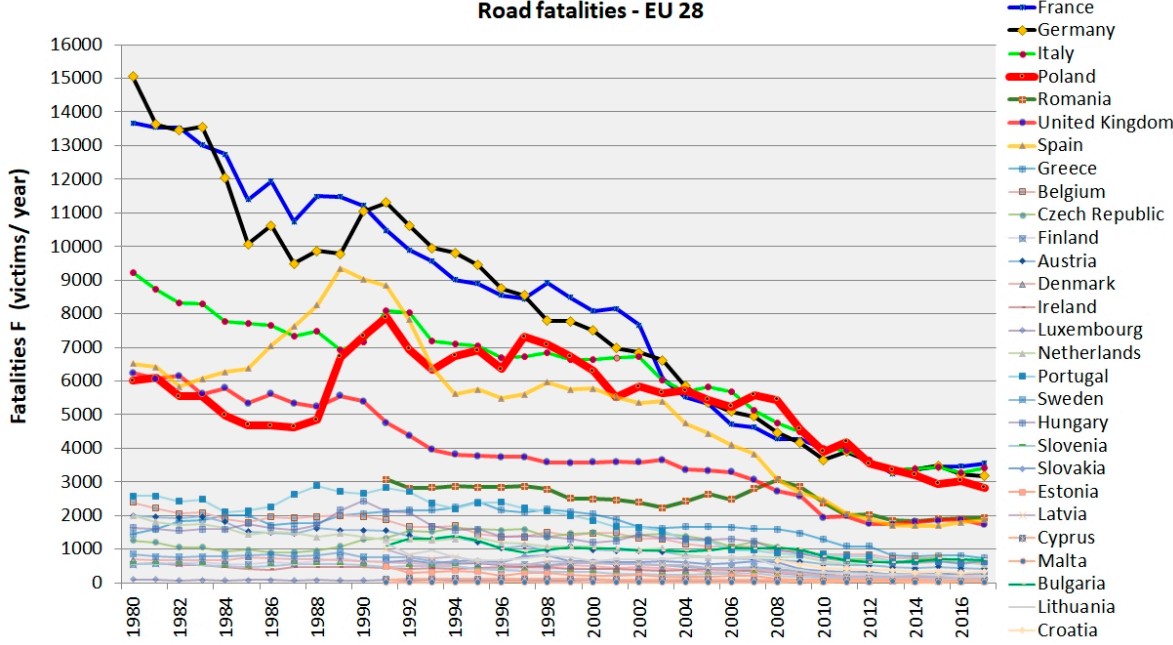

**Figure 2.** The changes in road fatalities in the EU-28 between 1990–2017.

Despite its efforts, the last fifteen years have seen Poland take the number one position among the European Union (EU) countries several times, ranking as the EU's most dangerous country. It is still one of the worst performing countries (today at number four). While fatalities are systematically

decreasing, the results are below the expectations. Still, just as it says in the motto of the Gdansk University of Technology "History is wisdom, future is challenge", Poland's road safety experience over the last three decades with its ups and downs does offer some wisdom, an invaluable asset in the face of future challenges of an ambitious road safety vision.

Produced by international organisations (World Bank, International Road Federation, and OECD), periodical reports show that road safety in developing Asian and African countries is much worse than in European countries just as Eastern and Central European countries have safety rates below those in Western and Northern Europe. As we know from numerous research papers [2–7,11–15], road safety is primarily determined by the level of a country's socioeconomic development and the mobility of its population. They are not, however, the only important factors [8].

The experience of OECD countries [16] shows that major improvements to road safety can be achieved through comprehensive and coordinated efforts and that success is likely if road safety programmes are well-prepared and implemented in a consistent manner. The Global Decade of Action for Road Safety is slowly coming to an end and the European Commission is drafting a new 5th EU Road Safety Action Programme 2021–2030. Other national road safety programmes for the next decade are under construction, including Poland's 5th Road Safety Programme (5th NRSP PL-2030). To ensure that safety programmes are well-prepared and that actions are effective and produce safety results, tools are needed to help with taking strategic decisions. Designed as strategic, with far-reaching visions and plans for new road infrastructure, road safety programmes must rely on road safety forecasts, road safety assessments and an understanding of how the planned actions will change road safety long-term (over periods of 10, 20 and even 30 years).

Experience shows that road fatality reduction efforts are most effective if they are part of focused programmes with clear targets and well-defined treatments and partners are committed to delivering on the targets and secured sources of funding. Model examples include Sweden's Vision Zero [17], The Netherlands' Sustainable Transport Safety [18,19] and Australia's Safe System [20–22]. Poland, just as many other developing Central and Eastern European countries (CEE), has gone through some turbulent social and economic changes in the last three decades, leaving their mark on changing road safety levels.

The objective of the article is to show the Polish transition from traditional to systemic approaches to road safety; to identify the main factors that helped to reduce road deaths; to evaluate the effectiveness of different approaches to road safety programming; to point out the successes and failures; to adopt a method for forecasting fatalities as a tool to draft road safety programmes; and to assess Poland's capacity to implement Vision Zero, to suggest actions to support the implementation of the vision and to formulate recommendations for developing countries. With this as the context, the following are the article's basic research (RQ) and practical (PQ) questions:

- RQ1. What are the approaches to road safety programming that can be recognised as effective?
- RQ2. Is it possible to adopt a general concept of how a country's road safety changes depending on its level of socioeconomic development?
- RQ3. What factors should be considered when developing long-term road safety forecasts for the purposes of strategic road safety programming?
- RQ4. What methods should be used in the strategic analysis of a country's road transport system development in the aspect of road safety?
- PQ1. What strategic goals can be adopted in selected reference periods (2030 and 2050) for further road safety programming in Poland?
- PQ2. Which groups of strategic actions should be taken to achieve strategic goals?

The answers are given in the individual sections and are a result of literature studies, comparative analyses, case studies, empirical and simulation studies and scenario analyses. Section 2 presents an overview of the road safety approaches and concepts and how they have evolved worldwide. Section 3 focuses on the Polish approach to road safety and a gradual transformation from the traditional to

systemic approaches. It presents proprietary tools for forecasting fatalities and for assessing road safety used to analyse the effects of socioeconomic changes on road fatalities and to assess the potential for achieving road safety targets as adopted. Section 4 gives a synthetic overview of the results achieved using Poland's approach to road safety and the effects of road safety programmes and actions on the levels of road safety. Section 5 builds on the methods presented in Section 3 and discusses the possibility of achieving Vision Zero until 2050 and successive (4th and 5th) national road safety programmes in Poland. Possible actions are identified to strengthen the capacity for Vision Zero implementation. Section 6 discusses the issues presented. Section 7 contains conclusions. Poland's experience, its successes and failures, may be useful to other middle-income countries which, just like Poland, are working to achieve safe road transport systems.

## 2. Overview of Road Safety Concepts

Because road accidents are a complex phenomenon, the last few decades have seen the emergence of numerous road safety concepts worldwide, a result of different perceptions of road accidents and how they happen. In the early 20th century, accidents were treated as random events and an unavoidable consequence of growing motorisation. This changed later when road users were blamed for accidents and legislative, educational and enforcement actions had to be used to change their behaviour. That was the basis of the traditional approach to road safety. An important milestone in the approaches to road safety was the emergence of the 3 Es concept (Engineering, Enforcement, and Education), followed by the 4 Es (with Emergency as the 4th E) and even the 6 Es (with Encouragement and Economy as the 5th and 6th Es) [23,24] defining the components of road safety prevention. The second half of the 20th century took a broader perspective on road accidents by recognising that they are a result of a combination of different events. That was the foundation of the systemic approach to road safety. Today, safety is seen as an integral part of state transport policy with tried-and-tested strategies, effective treatments and new approaches. The parameters of the transport system are determined by people's psychophysical capacities with a focus on sustainable transport and a commitment to zero road deaths.

There are two fundamental trends in how road safety is treated: the traditional and systemic approach (Figure 3, Table 1). With human error being the main cause of road accidents [25], the traditional approach claims that road users are solely responsible for road accidents giving it the name of the "road-user approach" [26]. Countermeasures are aimed at altering the road-user behaviour and are delivered in a bottom-up approach in response to the problems as they emerge (reactive approach). While single interventions help to improve road safety, they are hardly effective [27]. To respond to this and to address the flaws of the traditional system, a new approach evolved over the years, i.e., the systemic approach. Defined as holistic, the new approach looks at all subsystems of road transport, integrates road safety with other areas (such as the environment and public health) and promotes a shared responsibility for road safety. The responsibility is also to be shared by designers, road authorities, car manufacturers, etc., all of whom should ensure that human error (which everyone makes and will continue to make) can be forgiven and not lead to death or serious injury as a result of a road accident. According to the World Resources Institute (WRI) [27], the systemic approach to road safety was proved to be more effective in reducing traffic deaths and serious injuries when compared to traditional approaches. The implementation of safe system principles can help to meet broader goals towards sustainability in general. It is, in fact, a shift in the road safety paradigm (Table 1).

**Table 1.** A comparison of the traditional and systemic approaches to road safety.

| Characteristics | Traditional Approach | Systemic Approach |
| --- | --- | --- |
| Responsibility | Individual road user | Shared responsibility |
| Cause | Human error | System gaps |
| Actions | Reactive | Proactive |
| Interventions | Isolated | Integrated |
| Aim | Prevent road crashes and reduce casualties | Eliminate fatalities and serious injuries |

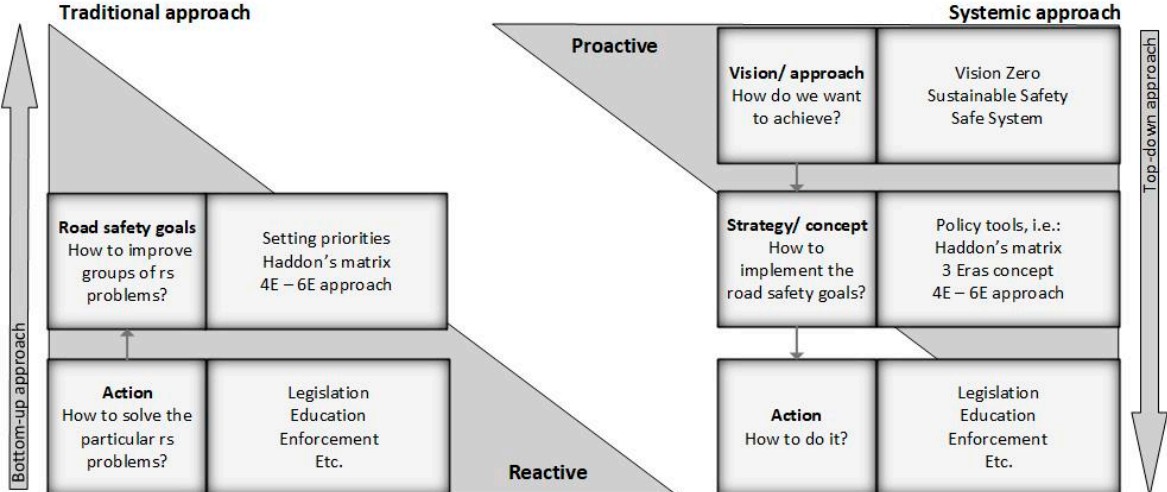

**Figure 3.** The traditional vs. systemic approaches to road safety.

The first to introduce the systemic approach were countries leading in road safety: Sweden and the Netherlands as early as the 1990s. The road safety effects were noteworthy, and many countries, in particular mature and high-income countries, followed suit and gradually adopted the approach [27]. The best known brand is Vision Zero [17,26]. Adopted by Sweden's Parliament in 1997, this philosophy aims to achieve, in a long-term approach, a state of the road transport system which does not involve road-user death or serious injury as a result of a road accident. The idea was picked up by a number of countries worldwide. The Vision Zero model is about the coexistence of vehicles with high NCAP (New Car Assessment Programme) scores, four star roads at a minimum (according to the EuroRAP programme) [28] and educated drivers complying with traffic regulations [29]. A competitive vision was formed in the Netherlands with its Sustainable Safety approach [18,19]. The idea is to eliminate or at least significantly reduce serious injury in road accidents by applying a systemic approach to designing and adapting the road transport system. In 2004, Australia adopted a combination of the two approaches in a sense. Called Safe System, the approach changes the perception and management of road safety [20–22]. It aims to maximise the benefits of mobility and to minimise the costs and consequences of road accidents. A founding principle here is that road accidents cannot be seen as a cost of growing mobility. While some literature sources (e.g., Reference [30]) present the Safe System as a way to implement Vision Zero or the Sustainable Safety philosophy, Australian sources [20–22] treat the approach as equal to the other concepts.

Unlike the traditional approach, the systemic approach is delivered top-bottom and has a long-term road safety vision. It is delivered at the strategic level using "policy tools", a term used in the literature [24]. It has room for concepts that developed over the years: 4E and its successive forms of 5E and 6E [23,24], Haddon matrix [31] and the 3 Eras concept [32]. An effective road safety strategy cannot just be written, it must be implemented. In the next layer of the systemic approach (Figure 3), steps are taken to identify hazards, to carry out preventative efforts to stop road accidents from happening and to provide rescue when a road accident happens despite all the efforts.

Figure 3 shows a clear contrast between the traditional and systemic approach to road safety. Both have left their mark on the Polish approach to road safety as explained in Section 3.

## 3. Methodology—Towards Safer Roads in Poland

### 3.1. Methodology of Analysis

For the purposes of this article, the following methodology is adopted. Having identified the concepts, policies and approaches to road safety (Section 2), the focus now is on analysing Poland's approach to fatality reduction and how it has evolved (Section 3.2). There is an identification of factors

that have an effect on road deaths with the critical ones used in our method for forecasting fatalities at the national level (Section 3.3). This methodology basis is applied further in the article.

*3.2. Polish Approach to Road Safety*

Poland's experience of road safety treatments is not very long. After 1991, the country's most tragic year (7,900 killed), a group of World Bank experts developed a report highlighting the high risks Polish road users were exposed to and the fact that there were no structures in place to address this and to take effective steps to improve the situation [33]. In 1993, following up on the report, the government appointed the National Road Safety Council (NRSC) and commissioned the first road safety strategy. This was, in fact, the start of Poland's systemic efforts for road safety. Table 2 presents the main ideas of the preventive programmes as they developed over the years. The first four National Road Safety Programmes (NRSP) are called GAMBIT and were written by a team headed by the Gdansk University of Technology.

**Table 2.** The main assumptions of Poland's national strategies from 1996–2020.

| Programme | | Road Safety Policy | Strategies | Actions |
|---|---|---|---|---|
| **Working Number** | **Name** | | | |
| I NRSP | Integrated Road Safety Programme GAMBIT'96 | None | Main qualitative goal, overall fatality reduction | Grouped (integrated) |
| II NRSP | Road Safety Programme for Poland 2001–2010 GAMBIT 2000 | None | Main target (4000 fatalities in 2010), 2 objectives | Two groups of tasks |
| III NRSP | National Road Safety Programme 2005–2007–2013 GAMBIT 2005 | Vision Zero | Main target (2800 fatalities in 2013), 5 strategic objectives, Operational Programme | 4E and system development |
| IIIa | Road Safety Programme 2007–2013 GAMBIT National Roads | Vision Zero | Main target (500 fatalities on national roads in 2013); Priorities, Pilot Programme | 3 Eras, 4E |
| IV NRSP | National Road Safety Programme until 2020 | Vision Zero | Main targets (2000 fatalities and 6900 serious injuries in 2020), 5 pillars | Safe System, 4E |

**I NRSP—GAMBIT'96.** Commissioned by the Minister of Transport and Maritime Economy, Poland's first comprehensive Integrated Road Safety Programme, called GAMBIT 96, was developed between 1993–1996. It did not have a specific quantitative target and there was one general goal which was to improve Poland's road safety. Its biggest advantage was that it planned to bring together sectors and industries around a shared task. Once established, the multidisciplinary team combined actors from different specialisations. It was the starting point of a long-term cooperation of a variety of stakeholders and the foundation of Poland's systemic approach. Thanks to GAMBIT'96, for the first time in history, the sectors of education, infrastructure, enforcement and rescue joined forces.

**II NRSP—GAMBIT 2000.** In 1999, a new structure was introduced dividing Poland's administration into four levels of governance: central, regional, county and municipal. The road network structure changed as well to incorporate the new setup leading the transport minister to commission a new version of the road safety programme. Called GAMBIT 2000, the programme formulated two groups of tasks: group A—the systemic work, including safety management, database and knowledge base creation, safety audits and training for professionals—and group B—covering the most critical problems and risks. The following were identified as Poland's main road safety problems: speeding, vulnerable road users, accident severity, transit roads passing through small towns and high-risk sites.

**III NRSP—GAMBIT 2005.** With the 2004 accession to the European Union, Poland was required to adapt its road safety programme to new conditions under the EU's transport policy and strategy set out in the White Paper and the 3rd Road Safety Action Programme with its goal to halve fatalities between 2000–2010 [34]. There were three time perspectives in the Programme: a vision of road safety until 2025 and beyond (based on Vision Zero), a road safety strategy until 2013 and an operational programme for the years 2005–2007. To achieve these targets, five strategic goals were formulated: (1) to build a basis for an effective and long-term road safety policy, (2) to develop safe road user behaviour, (3) to protect pedestrians, children and cyclists, (4) to build and maintain a safe road infrastructure and (5) to reduce accident severity and accident consequences. Each goal was to be achieved through actions divided into specific tasks. The programme assumed that a road safety system would be built with the necessary structures in place: education, enforcement, road infrastructure and rescue. The main idea was to integrate the five areas at three levels: national, regional and local (county and city) [35].

**GAMBIT National Roads.** Developed in 2007, GAMBIT National Roads was a sectoral programme for the entire network of national roads with a focus on infrastructure. It was also the first time for the General Directorate for National Roads and Motorways (the GDDKiA) to appreciate the role of other partners (teachers, journalists, police officers and fire fighters) all joining forces to improve road safety. This was, in fact, the basis for integrated actions under the 4E concept. The programme also set six fatality reduction objectives with deaths caused by hitting a pedestrian, head-on collisions, side and rear collisions, run-off-road accidents and fatalities occurring during nighttime and as a result of excessive speed. The tasks were organised into three groups following the 3 Eras concept (infrastructure, safety management and safety culture). The programme included a pilot action on national road no. 8 with tests and treatments to improve road-user safety. As a result, a sizeable fatality reduction was achieved of 30% after the road was treated.

**IV NRSP-2020.** The current National Road Safety Programme 2013–2020 builds on Vision Zero set out in the previous national road safety programmes [36]. As well as addressing the problem of road fatalities, the programme focus is on serious injuries. It aims primarily to halve the number of killed on Polish roads; to reduce serious injuries by 40%; to tackle speeding; and to improve the safety of pedestrians, cyclists and motorcyclists. Developed around the Safe System, the Programme looks at five pillars of action: safe people, safe road, safe speed, safe vehicle, medical rescue and postaccident care (consistent with the suggestions of the UN Decade). Each pillar sets priority actions to represent Poland's main road safety problems and the conditions for implementation. Each priority is a set of actions which encompass engineering—understood as technical solutions, enforcement—understood as enforcement and control—and education—understood as raising road safety awareness by learning about risk and understanding it. The programme also described actions to form part of rescue (4E).

*3.3. Method for Estimating Fatalities in Poland*

As work on national and regional road safety programmes commenced, it became clear that the necessary tools were simply not there and that scientific support was going to be needed to develop those tools. There was a lack of methods for

- long-term fatality forecasts nationally and regionally,
- assessing the effectiveness of proposed measures,
- selecting effective measures, and
- monitoring measures as they are implemented.

Given the lack of fatality forecast methods nationally and regionally, there were attempts to use other available methods and models [37], external experts' work [38] or simplified methods. But because international forecast methods did not represent the Polish conditions and simplified trend analyses left out a number of important factors, the fatality forecasts departed significantly from

the real figures. As a result, work began to develop Polish methods for forecasting fatalities at the national [8] and regional levels [39].

To assess safety at the national level (strategic), a risk-based approach was applied. Societal risk in strategic terms refers to all road traffic behaviour of entire groups in a selected area (country and region). The consequences (casualties and costs) are caused by road accidents within an agreed time interval (usually a year) which may occur as a result of hazardous events caused by the operation of the road transport system. The risk changes slowly and is influenced by the country's changing economy, social changes, better education, etc. Depending on the measure representing a specific area we can distinguish

- the overall risk calculated as the nominal consequence of accidents (number of victims and accident costs) and
- the normalised risk calculated as the total consequence per population in a given area, number of vehicles registered in a given area, gross national product, length of the road network and miles travelled.

The estimation of societal risk was conducted using a model with two components in which societal risk RS is taken as the quotient of a selected type of risk exposure E and a selected category of average consequence $K_{(E)}$ in relation to a unit of risk exposure on the road network of a country in a time period [8,40]. The model is described with Equation (1):

$$RS = E \times K_{(E)} \tag{1}$$

Two measures were used to assess societal strategic risk: the number of fatalities F as an overall measure and the road fatality rate dependent on demography RFR as a normalised measure making country road safety comparisons possible. The measures of societal risk F and RFR have a relation in common as shown below in Equation (2):

$$F = P \times RFR_{(P)} \tag{2}$$

The number of fatalities will be the quotient of the population P (population) of a country as a measure of risk exposure and the road fatality rate dependent on demography $RFR_{(P)}$ as the measure of the probability of road accident consequences which, in this case, means road deaths. $RFR_{(P)}$ is calculated (Equation (3)) as the quotient of the base $RFR_{(P),b}$ and modification factor MF taking into account the estimation error $\varepsilon_{(t)}$.

$$RFR_{(P)} = RFR_{(P),b} \times MF + \varepsilon_{(t)} \tag{3}$$

The $RFR_{(P),b}$ is calculated using a function dependent on the country's level of socioeconomic development (measured with gross national product per capita, GDPPC) and miles travelled (measured with the average distance covered by vehicles per capita, VTKPC). The rate represents the road safety performance function (RSPF) as exemplified in References [41–44]. The function is used in road safety management to identify the potential of treatments, to diagnose road safety problems, to identify high-risk sites and to assess the effectiveness of improvement countermeasures.

$RFR_{(P),b}$ is described with the power exponential function (Equation (4)) whose parameters were selected using multidimensional non-linear regression. This function was developed on a data set from 13 selected countries worldwide (having model characteristics) over the last 50 years (Figure 4).

$$RFR_{(P),b} = \beta_o GDPPC^{\beta_1} \times VTKPC^{\beta_2} \times \exp(-\beta_3 GDPPC + \beta_4) \tag{4}$$

**GDPPC**, the gross national product per capita (PPP, constant 2005, international $), is the relation between the national income and the number of population and increases together with the country's

socioeconomic development. When GDPPC increases, road fatality RFR increases at first following the power function until it reaches its maximum (at breakpoint BP for GDPPC = 14.7 thou. ID/inhab.), and after it has hit the breakpoint, it decreases following the exponential function.

**VTKPC**, the average vehicle kilometres travelled per year per capita, is the relation between the vehicle kilometres travelled and the country's population. The rate increases together with the country's economic growth asymptotically towards saturation which may differ from country to country depending on its transport policy, size and land use. As the VTKPC increases, the RFR increases following the exponential function. Figure 4 shows the diagrams of how the RFR changes in relation to GDPPC and VTKPC for average values for the other parameters which confirms the significant effect VTKPC has on RFR.

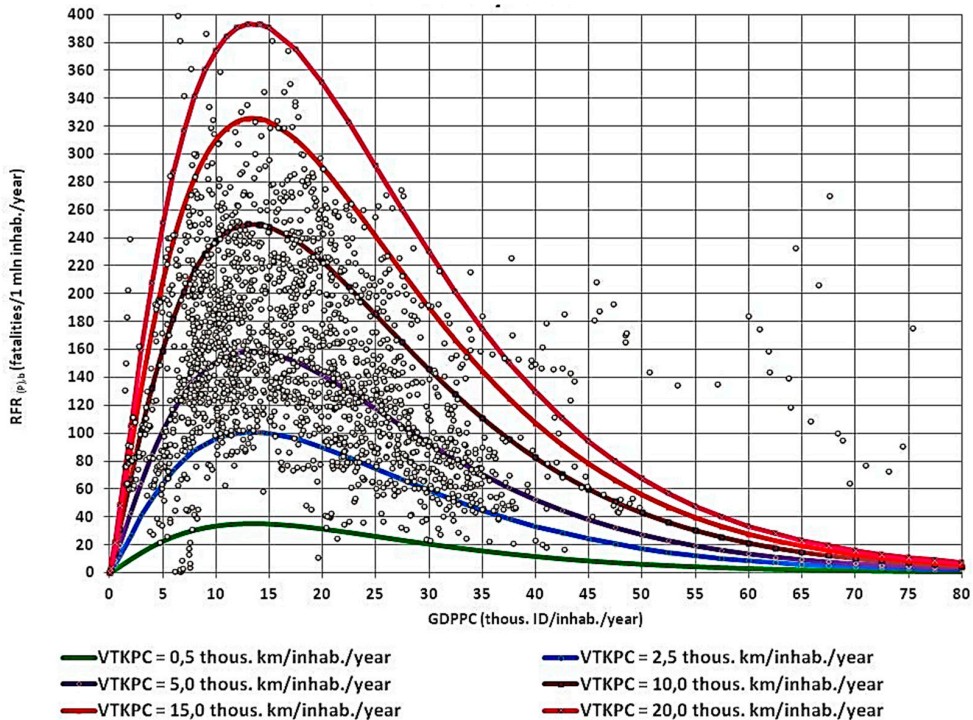

**Figure 4.** The changes in RFRb in relation to the Gross Domestic Product per capita (GDPPC) and the average vehicle kilometres travelled per year per capita (VTKPC) against real data for selected 13 countries (baseline data).

The modification factor MF is calculated as the relation between the real road fatality rate $RFR_{(P)}$ and the baseline $RFR_{(P),b}$ and calculated using the function in Equation (5). The modification factor $MF_b$ is calculated using the function described in Equation (6), which the independent variables of are parameters that are the most significant sources of risk (i.e., demographic, territorial, economic, social, motorisation and infrastructure factors).

$$MF = MF_b \times MF_c \tag{5}$$

$MF_b$ parameters were defined in the form of functions for select most relevant independent variables. The selection was made using a multidimensional non-linear regression based on the data from sixty selected countries worldwide from the last fifty years. Equation (6) presents one of the best matched models:

$$MF_b = GDPPC^{\gamma_1} \times \exp(-\gamma_2 LEI - \gamma_3 EDI - \gamma_4 CPI + \gamma_5 ACPC + \gamma_6 DPR \\ -\gamma_7 DME - \gamma_8 USB + \gamma_0) \tag{6}$$

As we can see from the analyses, the factors that are most relevant for a changing baseline $MF_b$ which modifies road fatality rates in the analysed countries are as follows:

- a decrease in RFR largely depends on GDPPC (in a range of > 14.7 tho. ID/inhab.), LEI, EDI, DME, CPI and USB and
- an increase in RFR largely depends on GDPPC (in a range of < 14.7 tho. ID/inhab.), ACPC and DRP.

**LEI**—the life expectancy index is the average number of years of life remaining at the birth year in the analysed country [45] and represents the country's health condition and quality of the health care system. This rate changes from 0 to 1. When LEI increases, RFR decreases following the exponential function.

**EDI**—the education development index is calculated from the overall gross education index for all levels of education [45]. It changes in a range of 0–1. As EDI increases, RFR decreases following the exponential function.

**CPI**—the corruption perception index is calculated by Transparency International [46]. It changes from 1 to 10; when CPI increases, corruption decreases: The higher the social development, the lower the corruption. When CPI increases (i.e., corruption decreases), RFR decreases following the exponential function.

**ACPC**—the alcohol consumption per capita is the average number of consumed alcohol per capita [25]. The level of alcohol consumption depends primarily on religious and cultural factors. When ACPC increases, RFR increases following the exponential function.

**DRP**—the density of paved roads is the relation between the paved roads and the area of the country. It increases together with the country's economic growth (very quickly at first, then slowly asymptotically towards saturation). When DRP increases, RFR increases following the exponential function. This is due to the higher speeds on improved roads.

**USB**—the use of seat belts by drivers and car occupants is calculated as a percentage share of road users using seat belts in a given country [47]. It changes in a range of 0–100. As USB increases, RFR decreases following the exponential function.

**DME**—the density of motorways and expressways is the relation between motorways and expressways and the area of the country. This rate increases together with the country's socioeconomic development when a certain income threshold is exceeded. It grows asymptotically towards saturation. When DME increases, RFR decreases following the exponential function (Figure 5a). This is due to high volume traffic moving to high quality roads.

The modification factor $MF_c$ is used to match calculation results to the conditions of a country. In the case of Poland, using data from the last twenty years, the rate was shown as a model described with Equation (7):

$$MF_{c,PL} = GDPPC^{\delta_1} \times \exp(-\delta_2 LEI - \delta_3 DEM - \delta_4 \ln(FV) + \delta_0) \tag{7}$$

**FV**—the number of speed cameras installed on a road network [48]: As that number increases, RFR decreases following the exponential function (quite rapidly at first) (Figure 5b).

Using the results of the analysis, a simplified and user-friendly version was developed for all groups of users (decision-makers, students, and journalists) presenting a method for estimating measures of societal risk (RFR and F) [9].

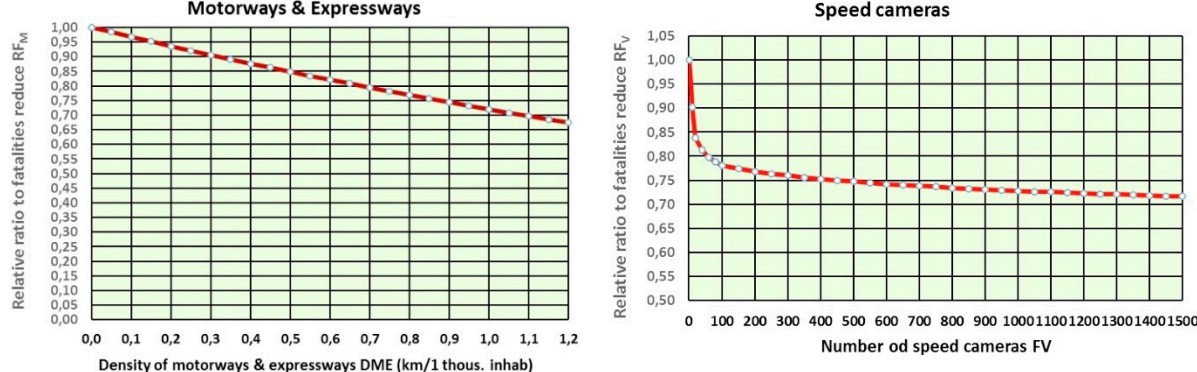

**Figure 5.** Charts showing (**a**) the effects of motorway and express road density on RFR and (**b**) the effects of the number of speed cameras on the level of fatality reduction.

## 4. Assessing the Effects of NRSP Implementation in Poland

### 4.1. Road Safety in Poland—Past and Present

The last thirty years are analysed. Initially, there were much fewer fatalities F in Poland than in France, Germany, Italy and Spain (Figure 2), and the road fatality rate dependent on demography RFR was much lower than in Portugal, Greece, Slovenia and Lithuania. As the years progressed, fatalities soared, causing Poland to top fatality F and road fatality rate dependent on demography RFR ranking lists for many years. Road safety efforts were launched and helped to reduce the number of killed significantly with Poland's fatality rate moving closer to that of the EU average (Figure 6). Today, with its 2800 fatalities in 2017, Poland takes fourteenth place among OECD countries and the fourth position in the EU. Four countries (France, Italy, Germany and Poland) had a combined rate of more than 50% of all fatalities in the EU in 2017. In the case of the road fatality rate RFR, Poland is also in the fourth place after Bulgaria, Romania and Latvia. What we know from analyses is that Poland has one of the EU's highest fatality numbers and road fatality rates. At present, Central and Eastern European countries, including Poland, have a rate of 75 fatalities per 1 million population, whilst Western European countries reach 35, which is twice as low. This shows the amount of work still to be done in this part of Europe, including Poland.

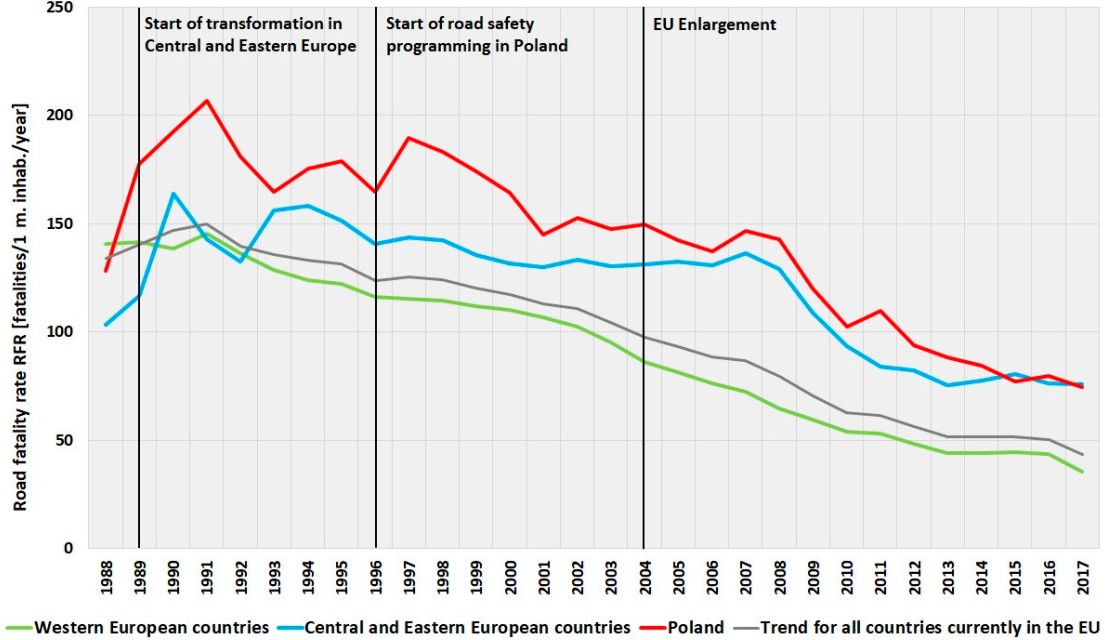

**Figure 6.** The road mortality rates in countries that are now EU members between 1988–2017.

In the analysis of the period from 1988 to 2017, fatality cut-off years are distinguished and selected characteristics are provided looking at demography, road traffic and safety. The values show how Poland's safety level has changed over thirty years. Within this period, the demographic rate $RFR_P$ dropped almost twice, the motorisation rate $RFR_M$. dropped almost seven times and the transport rate $RRF_T$ dropped nearly ten times.

## 4.2. Assessing the Progress of Road Safety Programmes

Since 1996, Poland has had four national road safety programmes. Tables 3 and 4 show the basic characteristics to highlight how the road safety level has improved during the programming periods. The results, especially looking at the changes in the dynamics of RFR decline, suggest that while the changes in RFR were a result of the changing socioeconomic situation, each road safety programme has also contributed to the fatality reduction and better road safety. Below is the characteristics of the key actions delivered under the programmes. Figure 7 presents those efforts, the NRSP's periods and the most important events related to road safety compared to Poland's fatality rates in the years 1986–2017.

**Table 3.** The characteristics of road transport in key years from 1988 to 2017.

| Year | Population | No. of Vehicles | Vehicle Travel Distance | No. of Fatalities | Road Fatality Rate | | |
|---|---|---|---|---|---|---|---|
| | P (mln) | V (mln) | VKT (b. vkm) | F (fatalities) | $RFR_P$ (fatalities/ 1 m. inhab.) | $RFR_M$ (fatalities/ 1 m. veh.) | $RRF_T$ (fatalities/ 1 b. vkm.) |
| 1988 | 37.8 | 6.9 | 113.2 | 4851 | 128.3 | 703.0 | 42.8 |
| 1991 | 38.2 | 8.6 | 108.3 | 7901 | 206.8 | 918.7 | 72.9 |
| 1997 | 38.6 | 12.3 | 127.4 | 7312 | 189.4 | 594.5 | 57.4 |
| 2001 | 38.2 | 14.7 | 148.4 | 5534 | 144.9 | 376.5 | 37.3 |
| 2007 | 38.1 | 19.5 | 220.8 | 5583 | 146.5 | 286.3 | 25.3 |
| 2015 | 38.0 | 27.4 | 3150 | 2938 | 77.3 | 107.2 | 9.3 |
| 2017 | 37.9 | 29.1 | 3300 | 2831 | 74.7 | 97.3 | 8.6 |

**Table 4.** The effectiveness of Poland's road safety programmes in the years 1996–2017.

| NRSP | Year of Programme End | Population | No. of Fatalities | Change in No. of Fatalities | Rate of Fatality Change | Percentage Drop in Fatalities | Road Fatality Rate |
|---|---|---|---|---|---|---|---|
| | | P (m.) | F (fatalities) | DF (fatalities) | TF (fatalities/year) | PF (%) | RFR (fatalities/ 1 m. inhab) |
| | 1995 | 38.6 | 6900 | - | - | - | 178.8 |
| I | 1999 | 38.7 | 6730 | −170 | −43 | −2.5 | 173.9 |
| II | 2004 | 38.2 | 5712 | −1018 | −204 | −15.1 | 149.5 |
| III | 2012 | 38.1 | 3540 | −2172 | −272 | −38.0 | 92.9 |
| IV | 2017 * | 37.9 | 2831 | −709 | −142 | −20.0 | 74.7 |

Note: * 2nd milestone of the IV NRSP

The previous period (1986–1995) covers the final years of communism and a planned economy and the initial years of Poland's political transformation. In the planned economy, Polish people could not access cars and those that were available were of lower grades. The country's bad economy limited access to petrol (which was rationed). As a result, mobility was low and people mainly used public transport to travel. Because of these limitations, fatality numbers were low. The problem began with political transformation (from socialism to democracy) and changes in the economy (from planned to capitalist). Following the introduction of the free enterprise act, people were able to buy cars (mostly second-hand cars bought west of the Polish border) and the number of cars on Polish roads soared. Young drivers with little experience of driving powerful and dynamic cars and the almost non-existent road police force (change of structure, staff and forms of operation) created an "explosive mix", and tragic events ensued on

Polish roads with fatalities almost doubled within two years (7901 killed in 1991). It came as a real shock to the government and the public. Commissioned in 1992, World Bank experts developed a report about the state of Poland's road safety. It lists the main problems such as a lack of adequate structures to lead road safety work. The report made it clear that road users in Poland are at a high risk [33]. Public shock, refusal to accept the risks to road users, media pressure and economic growth achieved within a short period of time helped to overcome the trend. The state of road safety in 1991 and the World Bank report provided the impetus for road safety programming. In the baseline year of 1995, there were 6900 fatalities.

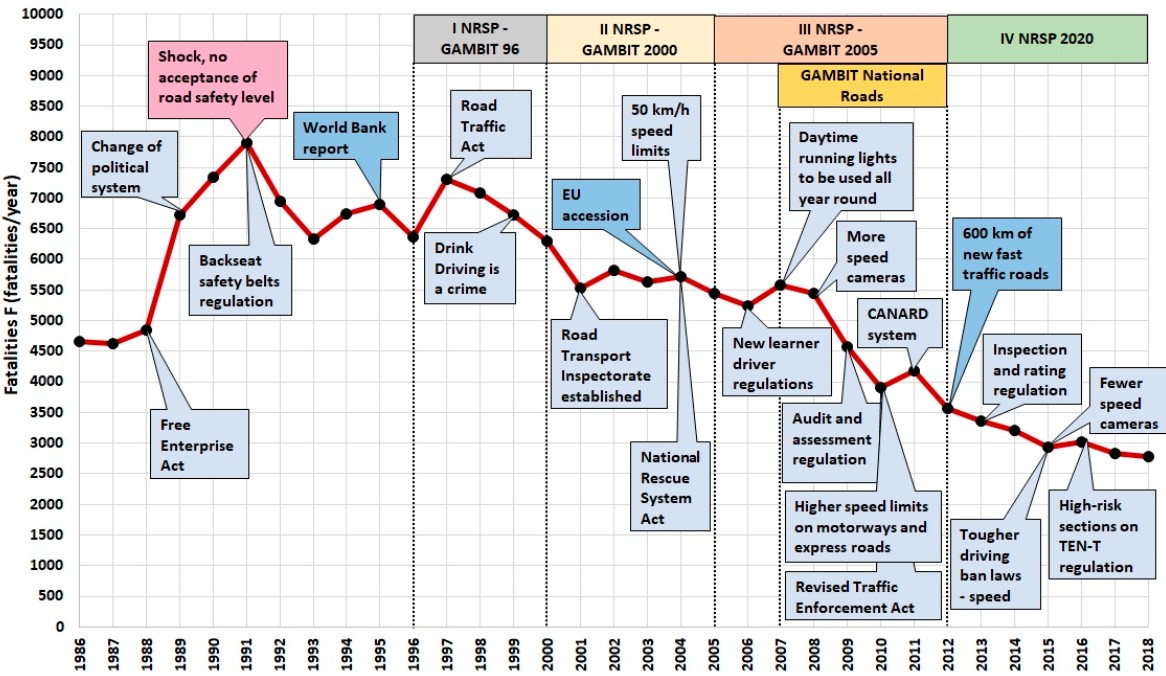

**Figure 7.** The fatalities in Poland against the background of events related to road safety.

**I NRSP—GAMBIT 96** (1996–1999) was a time of volatility for fatality trends in the years 1993–1997, a result of the central government putting the focus on priorities other than road safety. However, after 1997, there was a clear drop in fatalities which continued until 2001. During the programme period, there was a mere 2.5% fatality reduction down to 6730 victims (i.e., 170 victims less than in 1995). Implemented in 1997, the Road Traffic Act regulated and coordinated road traffic regulations and made sure that they were enforced. Its 1999 revision introduced druink driving rules making a blood alcohol limit above 0.05% a crime as opposed to an offence as was previously the case.

**II NRSP—GAMBIT 2000** (2000–2004). With the introduction of a new administrative division of the country into three tiers (regions, counties, municipalities), a new National Road Safety Programme GAMBIT 2000 had to be developed. An integrated approach to road safety problems was already incorporated. With a delivery timeline of ten years, the programme was, in fact, run for five years only. While it was in progress, fatalities went down by 15% to 5712 (i.e., 1018 less than in 1999). During this time, the Road Transport Inspectorate was established. Together with the police, the Inspectorate is also mandated with vehicle inspection and the control of haulage firms. While no immediate effect could be seen from the numerous actions in that time, long-term, they helped to achieve another major drop in fatalities in the years that followed. One such activity was Poland's membership in the EU, an external factor, which triggered new regulations and revised regulations designed to align Polish laws with those in other countries boasting much higher levels of safety.

**III NRSP—GAMBIT 2005** (2005–2012). With the accession to the European Union, a new National Road Safety Programme GAMBIT 2005 had to be developed. It spanned over nine years to match the EU's programming periods. While GAMBIT 2005 was in progress, a number of legislative, educational, preventive and infrastructural initiatives were taken at the national level. Only 84 of 144 tasks (58%)

were launched. While some of them did not produce the expected results or were poorly delivered, some of the political and administrative decisions taken at the time did not help either. Some measures, however, were good for road safety. They include the development and implementation of road safety programmes at the regional and county level in more than ten regions, cities and counties; the development and implementation of programmes for national roads and police; the start of work on building the Polish Road Safety Observatory and the organisation of two regional observatories; new learner driver training and exams; the implementation and development of an enforcement system (speed checks and driver working time control); the regulation of cycling on roads; the intensive construction of a network of express roads and motorways; the construction of safe junctions; the use of traffic calming measures; the introduction of road safety audits for some designs; the modernisation and development of the rescue system; and postaccident care.

The effects of GAMBIT 2005 could be seen in particular between 2007–2010 as new measures were introduced, such as the obligation to use daytime running lights all year round, deploying more speed cameras which helped to intensify enforcement and the implementation of tools recommended in the EU Directive of 2008 on road safety management (audit of the design documentation and assessing newly designed roads for their effect on road safety). Developed in 2007 and implemented in the years that followed, the first sectoral programme GAMBIT National Roads addressed primarily infrastructure work. "Roads of Trust" was another sectoral programme dedicated primarily to media campaigns to inform the public about road safety issues and to warn people about road hazards. The effects of EU road safety standards and principles started to show, aided by the growing influx of EU funding for building safe roads in Poland. During that time, the length of safe roads increased significantly (2012 set a record high number of 600 km of new motorways and express roads); new measures were introduced based on the EU Directive with the inspection of existing road infrastructure and the rating of high-risk sections. Other measures included a 50 km/h speed limit in built-up areas (sadly, 60 km/h during the night is still valid today) and new exams for learner drivers. It is estimated that within eight years of the programme, the measures helped to reduce fatalities by 38% to 3540 (i.e., less by 2172 compared to 2004), save about 6000 people from a fatal accident and generate savings as a result in the amount of some PLN 34.5 billion.

Polish [49] and international experts [50] believe that GAMBIT 2005 brought about a systematic fatality reduction. Polish experts are increasing their international activities, and there are more and more initiatives to improve road safety. The programme was instrumental in achieving that despite road accidents in Poland still not being considered an important problem, they are not on anyone's political agenda and the institutional effectiveness is low, a result of the shared (collective) responsibility approach to road safety problems. Sadly, many of the programme measures were never delivered. These include the failure to appoint a GAMBIT' 2005 leader; the structures of the country's road safety bodies, in particular the National Road Safety Council's structure, not having been improved; the initiative to appoint local lead bodies (inspectors, officers and leaders) was not followed through; road safety still does not have a well-operating funding system; strategies and their progress are not monitored properly; and successful road safety measures are not promoted. The year 2010 stands out as a bad example with motorway speed limits raised to 140 km/h and express road speed limits increased to 120 km/h. Another bad year was 2011 when the speed camera system underwent restructuring (the Police handed it over to the Road Transport Inspectorate) causing an increase in fatalities by 350 within a year. Analyses showed that Poland's road safety standards depart significantly from those in the European Union. The next national road safety programme took account of that.

**IV NRSP—2020** (2013–2020). Considering the flaws of how the previous programme was delivered and the success of the System Safety approach, in 2012, one more National Road Safety Programme was developed for the years 2013–2020 (valid today). The Programme includes a number of sectoral actions with a special focus on communication with the public, on the protection of vulnerable road users, on building safe roads and on the development of rescue and postaccident care. Five years into the Programme, fatalities are down by 20% to 2831 (i.e., a fatality reduction by 709 compared to

2012). The Programme has its flaws, such as downsizing the automatic speed camera system (in 2015) by taking down speed cameras from local authority roads. This may have influenced an increase in the number of fatalities by 150 victims in the following year. There is a real danger that the main targets will not be achieved because the fatality reduction in 2020 may be at 25–30% instead of the expected 50%, and the serious injury reduction in 2020 may be at 3–5% instead of the 40% target.

### 4.3. Examples of Successes and Failures

In summary, the period from 1988 to 2018 saw a threefold fatality reduction since the tragic 1991, Vision Zero is the new philosophy, systemic road safety measures have been adopted, regulations have been brought up to the level of countries with a good road safety record, road infrastructure has improved significantly, the technical quality and safety of vehicles on Polish roads has improved and Polish people are more aware of the importance of road safety. Unfortunately, there have also been road safety failures. They include speed limit regulations still in place today (60 km/h in built-up areas at night and 140 km/h and 120 k/h on motorways and express roads respectively); legislative flaws in the implementation of the automatic enforcement system; continuing shortages in the network of express roads and insufficient road safety standards on the remaining road infrastructure; a lack of road safety management tools for roads other than national roads (regional and local); dangerous road user behaviours, especially in the driver–pedestrian relation; driving over the speed limit; risky manoeuvres; and aggressive styles of driving. All thess keep fatality reduction at a slow pace, and Poland's road safety compared to the EU's best-performing countries is still too low. While there are successes to be proud of, the failures must incentivise us to do even more. With the work described in Section 4.2, Poland has made great strides towards better road safety, especially when it comes to the measures identified in the programmes. Figure 8 shows Poland's eight critical road safety problems generating serious risks of fatal accidents: national roads, pedestrians, dangerous speed, high severity, nighttime, "hard" roadsides, drinking and driving and intersections. An analysis of the problems shows that while fatalities in those cases have been significantly reduced, the numbers are clearly too high.

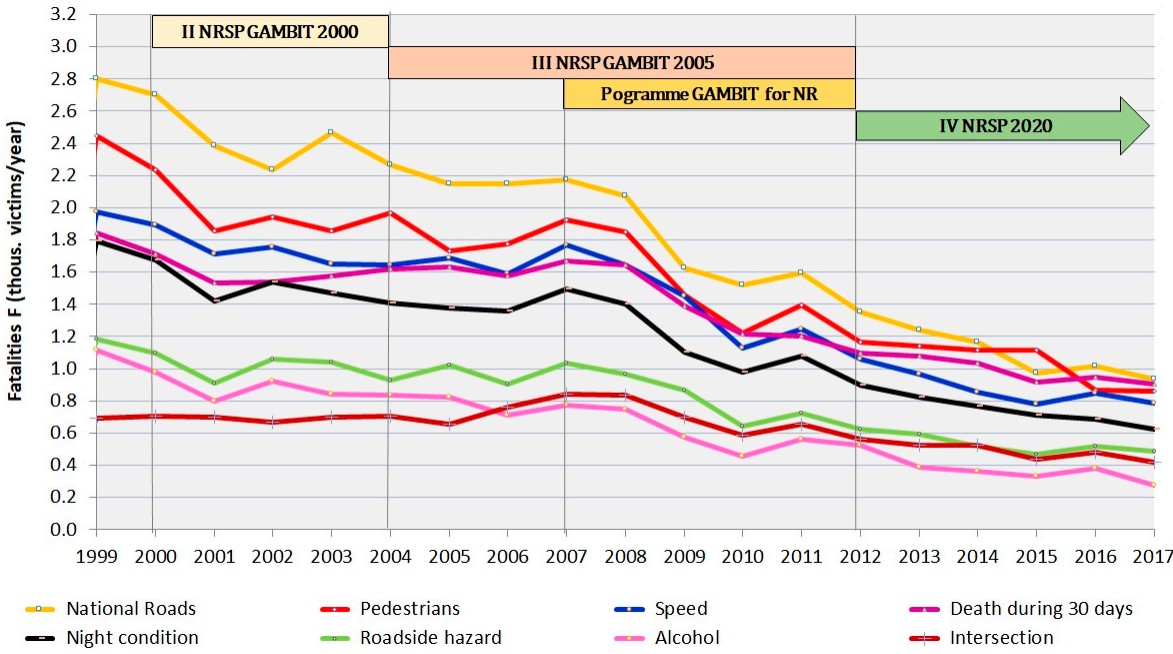

**Figure 8.** The changes in road accident fatalities between 2001–2017 in Poland broken by the problem areas.

**Speed**: If it is excessive, dangerous or not right for the conditions, speed is the cause of 31% of fatal accidents in 2001 and 27.8% in 2017. Within 16 years of the analysis, fatalities caused by speeding went

down by 54%. In this period, systematic measures were implemented, designed to prevent dangerous speeds (automatic speed camera system CANARD, traffic calming in many cities and the construction of safe roads) [51]. While mistakes have been made (downscaling the speed camera system and higher speed limits on motorways), fatal accidents caused by excessive speeds continue to drop.

**National roads** are managed by the GDDKiA and carry more than 25% of Poland's road traffic. Road accidents on these roads account for 19% of all accidents with fatalities representing as much as 33% of all fatalities. Within 16 years of the analysis, fatalities on national roads went down by 55%. In this period, a number of systemic actions were carried out: the development and implementation of a sectoral programme GAMBIT National Roads and the information campaign Roads of Trust, the continued development of fast traffic roads (between 2003–2016 more than 2500 km of new sections were built) and the continuous improvement of safety standards. The effects could have been better, but roads of high technical standards are still in short supply (motorways and express roads), many cities and towns do not have ring roads, some roads still have the wrong cross sections, roadsides do not meet technical and safety standards, there is a lack of facilities for vulnerable road users, road safety standards are not met when roads are resurfaced and ITS measures as part of road traffic management are still poorly represented [36].

**Pedestrians** are most at risk of serious injury or death in a road accident. Poland is the number one among the European Union's most dangerous countries for risks to pedestrians in road traffic (31% of all fatalities). For many years, about 2.1 thousand pedestrians were killed every year in Poland, and it was not until 2007 that a strong downward trend began and continued in the following years. By 2017, pedestrian fatalities dropped to 0.8 thousand (i.e., by 53% compared to 2001). The factors that increase the risk of pedestrian accidents include poorly regulated pedestrian priorities (work is already under way to change the law), the lack of pedestrian protection (pavements and pedestrian islands) and poor pedestrian visibility at night [52]. At the same time, there has been some progress, such as the manual for pedestrian traffic organisers [52] which became the basis for improving pedestrian infrastructure safety, especially at the local authority level. The driver–pedestrian relation has clearly improved, but more educational work is still needed. There are dedicated lanes for pedestrians and cyclists on rural sections of national and regional roads.

**Death within 30 days**: Road accident severity in Poland is high (10 fatalities per 100 accidents) and is the result of high speeds in conditions with a lack of lane separation and harsh roadsides, a deficient rescue system and problems in the health service. As a consequence, in 2017, as many as 32% of casualties died within 30 days of the accident compared to 28% in 2001. Within 16 years of the analysis, the number of deaths within 30 days from the accident dropped by 41% only. Sadly, none of the relevant GAMBIT 2005 targets have been met. Considering this, three areas should be addressed: reducing accident severity (through improved infrastructure, organisation and management), improving the road rescue system and improving postaccident care.

**Night-time**: Fatal accidents usually happen in the absence of daylight (night, dusk, and dawn) or due to fatigue or drivers nodding off. In 2017, as much as 45% of all fatalities died in these types of accidents. Within 16 years of the analysis, the number of people killed in road accidents caused by drivers during the night dropped by 42% only. One of the reasons for the slow decline is higher speeds during nighttime in built-up areas which continue to have a speed limit of 60 km/h from 12 am to 6 am. It is also common practice to switch off traffic lights during the night. The problem of rural roads is that road users, both pedestrians and drivers, have limited perception of the road. Other causes of fatal accidents include the condition of vertical and horizontal signs which do not meet reflectivity requirements; poorly lit roads, especially at sensitive sites such as junctions; and pedestrian crossings.

**Unsafe (hard) roadsides**: Run-off-road accidents continue to be of the most serious road safety problems. When they happen, they cause secondary crashes ending in vehicle rollover or hitting a roadside object. These accidents represent about 25% of rural accidents and nearly 16% of all fatalities in Poland. Within 16 years of the analysis, the number of people killed as a result of a roadside accident went down by 42% only. The main cause of this is that the regulations are not clear about roadside

design and maintenance; there are conflicts with environmentalists over the cutting of roadside trees which, again, is a result of bad legislation leaving decision-makers free to make their own decisions. Despite the criticism, national and regional road safety programmes are being followed with tree felling now a requirement for road projects, the use of structures that offer good containment and a new approach to safety barriers.

**Drinking and driving**: In the late 1990s, alcohol was one of the main road safety problems in Poland. The share of fatalities in drunk driving accidents was 22%. Within 16 years of the analysis, the number of people killed in drunk driving accidents dropped by as much as 65%. In 2017, drinking and driving accidents caused 10% of fatalities. In this respect, Poland tops the list as a country with the lowest share of fatal drinking and driving accidents. This is a result of a number of radical and systematic efforts by traffic enforcement services (police and Road Transport Inspectorate), by education and information and by a new culture of alcohol consumption in Poland.

**Intersection**: The problem of road intersections is connected above all with road infrastructure. Reducing the number of fatalities means using safer types of intersections (roundabouts and traffic lights), improving visibility, readability and passability within them. In 1999, the share of fatalities in accidents at crossroads was around 10%, while in 2017, it increased to 14% and the number of fatalities decreased by 40%. Further actions are necessary mainly in the field of building a modern, safe road infrastructure but also in activities aimed at the behaviour of drivers (speed and driving culture).

## 5. Assessment of the Implementation of Vision Zero in Poland

### 5.1. Method of Strategic Road Safety Planning

Strategic planning was proved to be effective as it provides a strong foundation for joint and effective actions to achieve the set objective [53,54]. The process defines a long-term goal and how it is to be achieved. It has two phases: strategic analysis (which looks at the environment, conditions and resources, determines the strategic position and formulating the basis for forecasts) and strategy design (determining the vision and strategic goals, developing and assessing strategic alternatives and selecting the strategic plan). In strategic planning, the emphasis is on determining the effects of the environment (creating opportunities and threats) on the implementation of strategic goals. It is also important to understand how important internal factors affect the operation of the subject of analysis which in this case is the road transport system.

In order to determine these impacts, strategic planning should meet the following conditions: focus on long (rather than short) time horizons, analyse structural change rather than simply extrapolate previous trends, expect an answer to the question "what if?" rather than wait for unconditional forecast results, use theoretical grounds and databases to estimate values of key variables and try to understand the future and not only predict it [55].

In strategic analyses, two methods are commonly used: the scenario and non-scenario approaches. Non-scenario methods use one variant of how the environment will develop as an extrapolation of previous trends. The methods use mathematical tools i.e., trend analysis, Delphi method, time series forecasting, econometric models, expert panels, "brainstorming" and Bayes method. In contrast, scenario methods help to identify factors affecting the analysed system and forecast potential changes in how these factors will be formed, all for alternative scenarios and multivariant options of implementation. Scenario methods help to reduce the uncertainty of how the analysed system will function in the future by creating various possible scenarios of future phenomena [56]. Scenario methods use different mathematical tools, i.e., scenarios of the likelihood of events, simulation methods, scenarios of the state and processes in the environment, as well as some of the tools used in non-scenario methods. Scenarios, as a basic technique for research and strategic analyses, have long been used by government planners, corporate managers and military analysts as powerful tools to support decision making in the face of uncertainty. Scenario methods are also used in transport planning, just as in the case of the RAND corporation and the Institute for Mobility Research, who prepared and analysed

scenarios for how the mobility of people can change in the US by 2030 [57]. The information obtained from the study can help to better prepare the transport system operation for the future.

Each method of strategic analysis uses methods for forecasting changes in key system variables. In practice, two approaches are used to forecast the performance of the analysed system: structural and non-structural. In the non-structural approach, we assume that the information on the mechanism forming the forecasting variable is contained in its past. Naïve and exponential smoothing methods are used, in which we assume a small random fluctuation of the analysed factor and no significant changes in the main factors forming the observed phenomena. The methods are characterised by simple model design; however, in order to ensure sufficient estimator accuracy, high requirements regarding sample size need to be met. The disadvantages of naive methods include a high impact of random variations on the forecast results, the inability to estimate the ex ante forecast error and a low accuracy in longer forecasting periods. In the second approach, the forecasting instrument is a structural model, which the equations of have been constructed to reflect a certain theory and constitute a description of the mechanism generating the forecasted values. If properly used, the adopted theory (economic, safety, etc.) brings additional information, a priori information in relation to the information contained in the statistical sample, and thus complements the not always abundant statistical information. In the structural approach, complex casual methods are widely used. These are based on the data on the causal factors and estimated causal relationships between the factors. Causal forecasting methods have a clearer value in planning and can be used in any planning phase. In practice, we use a mix of both approaches, structural and non-structural, and both should be regarded as complementary and not competitive. The selection of one of them is determined by different circumstances, i.e., the amount of available a priori information (adopting the form of theory) and the stock of available statistical information contained in the sample.

Taking into account these considerations and the imperfections of strategic road safety programming, a concept was proposed to adapt both the scenario and non-scenario method in the strategic planning of the road transport system development with regards to road safety (Figure 9).

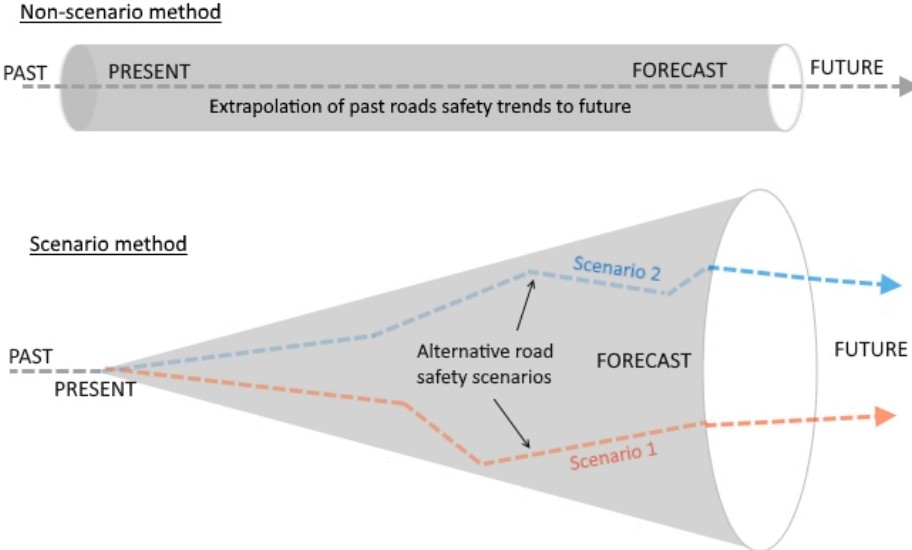

**Figure 9.** The concept of scenario and non-scenario methods applied to strategic road safety analysis. Source: own elaboration based on Reference [57].

Non-scenario methods were and are quite commonly used for road safety analyses, mainly to explain the causes of temporary changes in the trends of key variables and for short-term (1 to 3 years) strategic planning [58,59]. However, when developing road safety programmes, it is necessary to set long-term strategic goals and to select the right tools and resources to implement them. In this case, scenario methods should be used to a larger extent. The works of Koornstra [5] and Stipdong [60]

can serve as good examples of the application of the scenario method to the strategic analysis of road safety systems development. In 2007, Koornstra [5] used the method to forecast the number of fatalities in road accidents in eight countries between 2010 and 2050, adopting four scenarios: reference, learning and two alternative scenarios. The learning scenario was indicated as the most effective one [5]. Stipdong et al. [60], as part of the ASSESS research project, used the scenario method to examine the chances of EU countries to achieve the strategic objectives set out in the White Paper on transport [61]. Four scenarios were adopted that differed in the scope of activities carried out to improve the road transport system in the EU. As a result, it was stated that in order to achieve the objectives, all measures recommended in the White Paper and an additional set of measures had to be implemented [60].

### *5.2. Scenario Method for Road Safety Strategic Analysis*

### 5.2.1. Research Method

Most countries have their own development strategies that are full of ambitious goals. In Poland, over the past years, the requirement of strategic thinking has been put on the back burner. The primacy of the current image and short-term interests has become an overriding issue in managing the development of the country. The problem of a long-term strategy was largely left to the individual initiative of scientists and their passions. As a result, Poland lacks a long-term programming of conditions for future development. Activities carried out in many key areas are of an ad hoc nature. In the case of road safety management in Poland, as demonstrated in Section 4, strategic planning has been used for over 20 years initially, to a limited extent, setting only objectives and directions of actions and later defining the vision, the main objectives, detailed strategic objectives, priorities and directions for strategic activities, operational plans, necessary resources (organizational and financial), and how these will be implemented and monitored. In recent road safety programmes, scenario methods were used to determine the effectiveness of the proposed programmes, but the objectives were still adopted arbitrarily without a deep strategic analysis.

In the run-up to the next road safety programming period in Poland, prior to developing the next (V NRSP) road safety programme, a research task was undertaken designed to define the programme objectives and directions of strategic activities in two time horizons: 2030 and 2050. The study was conducted using the simplified scenario method described in Section 5.1, divided into two stages: (1) creating scenarios (the identification of objectives and research areas, trend analyses, the analysis of opportunities and threats and the development of scenarios for strategic activities) and (2) scenario assessments (conducting analyses and forecasting simulations; the assessment of the effectiveness of the proposed groups of activities under various social, economic and technical conditions; and recommendations to the programme).

### 5.2.2. Creating Scenarios

Prior to the analysis, the original research questions were brought back: they are research questions RQ3 and RQ4 and practical questions PQ1 and PQ2. Answers to these questions are important and can help Poland or other countries to prepare rational strategic goals and directions of strategic actions of new road safety programmes, taking into account opportunities and threats to their implementation.

**Vision and strategic goals.** Poland has been implementing the principles of Vision Zero constantly for many years, and it was assumed that this will be continued in the analysed period. It is different in the case of strategic goals. Considering the adopted principle of a 50% fatality reduction in the next decade, it can be initially assumed that the main objective for 2030 should be not more than 1000 fatalities and not more than 250 fatalities in 2050. In the second case, the goal is much above the expectations of the European Commission [61] for the year 2050. The answer to the practical questions (PQ1 and PQ2) cannot be reliably obtained with trend analysis or time series forecasting because data and information on long-term changes in significant demographic, social, economic and

transport factors are uncertain, incomplete, evolving or contradictory. Therefore, in order to answer the questions, the scenario method was used.

**Analysis of trends in factor changes, opportunities and threats.** Transport planners and decision makers have to make decisions in a specific, long time horizon (up to 30 years). Although we already know that the RFR decreases along with an increasing level of a country's socioeconomic development, determining the road safety level in the analysed period is still a serious challenge. In order to identify possible trends in the number of fatalities and the factors affecting these changes, previous research results were used [3,4,6–9]. For further analyses, a group of eleven variables was adopted (Table 5), describing Poland's road transport system (VTKPC, DPR, DME, USB and FV) and the external environment of the system (P, GDPPC, LEI, EDI, CPI and ACPC).

**Table 5.** The areas of influence and their elements, variables and projections.

| Area of Influence | Element | Variable | Projections |
|---|---|---|---|
| External environment | Demography | Population: P | Systematic reduction by 1.2–2.5 per mil annually |
| | Economy | Gross domestic product per capita: GDPPC | Average increase of 2.5% per year |
| | Health system | Life expectancy index: LEI | Systematic increase |
| | Education system | Education development index: EDI | Systematic increase |
| | Organisational system | Corruption perception index: CPI | Systematic increase |
| | Societal | Alcohol consumption per capita: ACPC | Maintaining a constant level |
| Road transport system | Travelled distance | Average vehicle kilometres travelled: VTKPC | Systematic increase of approx. 2% annually |
| | Road network | Density of paved roads: DRP | Systematic increase |
| | Network of safe roads: motorways and express roads | Density of motorways and express roads: DME | Sharp increase in the next EU funding period, followed by a minimum increase |
| | Road users behaviour | Use of seat belts by drivers and car occupants: USB | Systematic increase |
| | Automatic enforcement system | Number of speed cameras: FV | Large fluctuations, depending on the current government |

At the strategic level, the three most important factors affecting the level of mortality in road accidents are the number of inhabitants, the distance travelled by vehicles and the country's wealth (but mainly its part devoted to the development of a safe road transport system). Other factors should be considered as factors modifying the influence of the main factors.

An analysis of the projections indicates that there are chances to achieve the initial objectives because of the expected systematic economic growth, significant extension of a safe roads (motorways and express roads) network, use of new technologies, positive change in transport behaviour and safety culture and reduction (although undesirable) in the population. Unfortunately, there are also threats to achieving the objectives which may include a lower economic growth rate leading to a slower development of a safe road network, poor (not very decisive) transport policies in road safety, an increase in the share of car transport, a deteriorating or not developing enforcement system (caused by opportunism of politicians) and a negative transport behaviour of the public.

**Developing scenarios for strategic activities.** The assessment of the implementation of Vision Zero assumptions in Poland until 2050 and preliminary assumptions of the next 5th Road Safety

Programme in Poland by 2030 (V NRSP-2030) was carried out considering the conducted research (Section 3) and the studies, analyses and evaluation of the functioning of the four National Road Safety Programmes in Poland. The assessment was performed with the use of the scenario method. Four scenarios were prepared (summarised in Table 6) that contained groups of key strategic actions, including two groups of factors: the level of socioeconomic development, measured by GDP growth (very good/good and worse/bad levels), and the effectiveness of transport safety policy actions (higher/low levels).

**Table 6.** The potential road safety scenarios of Vision Zero realization in Poland.

| Factors Impacting on Scenarios | | Social and Economic Situation Measured with GDP | |
|---|---|---|---|
| | | **Very Good/Good** | **Worse/Bad** |
| Effectiveness of transport policy dependent on road safety | Higher | S.1 optimistic scenario | S.3 stagnation scenario |
| | Low | S.2 moderate scenario | S.4 pessimistic scenario |

Using the original authors' method of the long-term forecasting of the number of fatalities in road accidents (described in Section 3.2) and in References [2–7], the forecast was conducted on the number of fatalities for four scenarios for the development of the road safety system in Poland by 2050, listed in Table 6. In all scenarios, the same number of inhabitants was assumed, with the average rate of its reduction 0.1% per year (calculation based on data from the years 2000–2017). The base year for the analysed scenarios was 2017, in which 2831 people were killed on Polish roads, and the death rate in relation to the RFR = 75 fatalities/1 million inhabitants.

Table 7 presents the predicted values of the most important strategic variables which represent the changes in the demography (population P); the socioeconomic development (GDPPC); and the country's organisational systems, specifically the health system (LEI), education system (EDI), organisational system (CPI), road transport system (vehicle kilometres travelled VTK, length of motorways and express roads LME) and enforcement system (number of speed cameras FV). The numerical values of particular strategic variables (Table 7 and Figure 10) were estimated based on a mathematical analysis using own prognostic models [8] or taking additional assumptions.

**Table 7.** The predicted values of selected variables used to estimate the number of fatalities in particular scenarios of Vision Zero delivery in Poland.

| Strategic Variables | Unit | Year/Period | Scenario | | | |
|---|---|---|---|---|---|---|
| | | | **S1** | **S2** | **S3** | **S4** |
| Population | P | m.inhab./year | 2020 | 36.0 | 37.8 | 37.8 | 37.9 |
| | | | 2030 | 33.7 | 36.8 | 36.8 | 37.5 |
| | | | 2050 | 29.6 | 33.0 | 33.0 | 36.6 |
| Gross Domestic Product per capita | GDPPC | thous. ID/inhab./year | 2020 | 29.5 | 27.9 | 26.0 | 26.0 |
| | | | 2030 | 62.5 | 38.5 | 33.0 | 33.0 |
| | | | 2050 | 73.8 | 62.5 | 50.5 | 50.5 |
| Vehicle kilometres travelled | VTK | b. vkm/year | 2020 | 330.4 | 347.8 | 347.8 | 352.5 |
| | | | 2030 | 337.1 | 382.7 | 382.7 | 393.8 |
| | | | 2050 | 346.3 | 389.4 | 389.4 | 435.5 |
| Length of motorways and expressways | LME | thous. km/year | 2020 | 5.3 | 5.3 | 5.3 | 5.3 |
| | | | 2030 | 7.6 | 6.9 | 6.4 | 6.4 |
| | | | 2050 | 8.0 | 7.2 | 6.5 | 6.5 |
| Number of speed cameras | FV | items | 2020 | 400 | 400 | 250 | 250 |
| | | | 2030 | 930 | 600 | 280 | 280 |
| | | | 2050 | 1150 | 720 | 330 | 330 |

Analysing the variability of the adopted strategic variables, it can be stated that by 2050, in Poland, the following changes may occur depending on the analysed scenario:

- the population will decrease by 0.4–0.8%;
- the economic potential measured with GDPPC will increase by 110–215%;
- the country's organisational system will develop and the systematic increase in the LEI, EDI and CPI indices may be expected (Figure 10 presents the predicted changes in the indices; in order to fit the scale, the normalised CPIx was used: CPIx = 0.1·CPI);
- the road transport system will develop and the length of motorways and express roads LME will increase by 140–215%, which will contribute to an increase in VTK by 50–135%;
- the traffic enforcement system will improve and the number of speed cameras (FV) allowing for an automatic enforcement will increase by 35–475%; and
- road user behaviour will improve (less drunk driving and higher seatbelt usage).

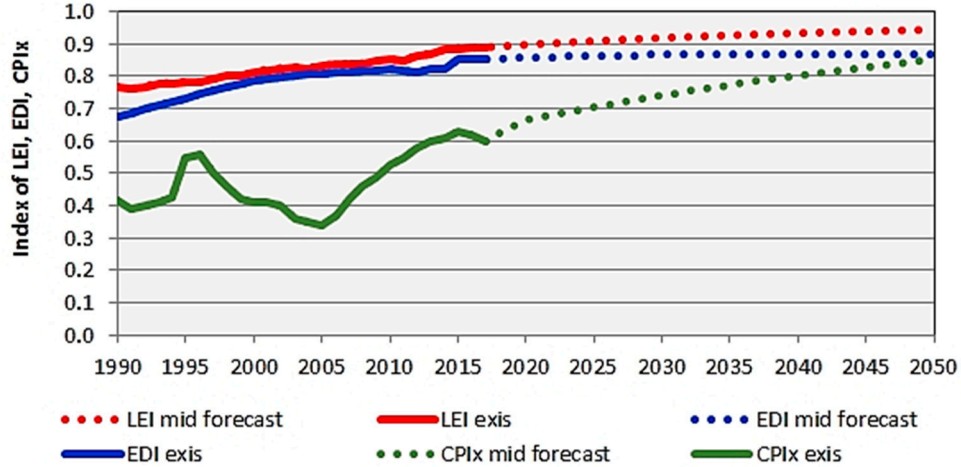

**Figure 10.** The reported and predicted changes in the values of LEI, EDI and CPIx indices in Poland by 2050.

Moreover, in individual scenarios, additional assumptions for changes to other parameters by 2030 were adopted with the possibility of maintaining a similar pace of change by 2050.

**S1 optimistic scenario**: (a) an increased rate of the country's economic development, i.e., a GDP growth above 3.5% per annum, which may result in: increased expenditures for the road network development (building modern and safe roads) and the modernisation of local roads (with a mandatory use of road safety standards), increased expenditures for health and rescue services on roads, etc. (b) A strong transport policy with regard to the road safety, the strengthening of the leading position, the strengthening of the role of road safety institutions, maintaining the pace of the construction of express roads and other roads with high road safety standards, the implementation of measures covered by the modified Directive on road infrastructure safety management, the maintenance and development of the traffic control system (increased number of speed cameras and sections with automatic speed supervision) and new traffic management systems (including the development of the national, regional and urban systems).

**S.2 moderate scenario**: (a) the country's further economic development, i.e., an increase in GDP about 3% per annum, which may result in: maintaining the existing or increasing expenditures for the development of a network of modern and safe roads, increased expenditures for health and the rescue on roads, etc. (b) A responsible transport policy with regard to the road safety, the strengthening of the leading position, the strengthening of the role of road safety institutions, the maintenance of the pace of construction of express roads and other roads with high road safety standards, the implementation of measures covered by the Directive on road infrastructure safety management, the maintenance and

development of traffic control system (a slightly increased number of speed cameras and sections with automatic speed supervision).

**S.3 scenario of stagnation**: (a) the country's slow economic development, i.e., a GDP growth below 3% per annum, which may be associated with limiting the increase in expenditure for the development of a network of modern and safe roads and a reduced expenditure for health care, road rescue, etc. (b) A poor transport policy with regard to the road safety, the lack of leadership, the limitation of the role of road safety institutions, the reduction of the pace of construction of express roads and other roads with high road safety standards, the delayed or limited implementation of measures covered by the modified Directive on road infrastructure safety management and the limitation of traffic control system operations (including a limited number of speed cameras and sections with automatic speed supervision).

**S.4 pessimistic scenario**: (a) the country's slow economic development, i.e., a GDP growth about 2% per annum, which may be associated with a decrease in expenditure for the development of a network of modern and safe roads and a reduced expenditure for health care, road rescue, etc. (b) A very poor transport policy with regard to the road safety, the lack of leadership, the limitation of the role of road safety institutions, the reduction of the pace of construction of express roads and other roads with high road safety standards, the lack of implementation of measures covered by the modified Directive on road infrastructure safety management and the limitation of traffic control system operations (including a limited number of speed cameras and sections with automatic speed supervision).

*5.3. Estimation of the Effects of the Individual Scenarios*

In order to assess the effects of individual scenarios, four measures were adopted, representing

- the level of road safety in the reference year (2020, 2030, 2050):

    1.  number of fatalities F (fatalities/year)
    2.  road fatality rate RFR (fatalities/1 m. inhab./year)

- the combined road safety effect resulting from the scenario in the analysis period (33 years):

    3.  expected, total number of fatalities SF (fatalities/33 years)
    4.  estimated number of people who can be saved from death (PRD) compared to the worst scenario S4 (fatalities/33 years)

The number of fatalities F and the road fatality rate RFR were estimated using the original authors' method of long-term forecasting of road accident fatalities (described in Section 3.3) and in Reference [2–7]. This was done for four scenarios of Poland's road safety system development until 2050, listed in Table 6 with the variables presented in Table 7. The other two variables (SF and PRD) were estimated using Equations (8) and (9).

$$SF_j = \sum_{i=1}^{33} F_{ij} \qquad (8)$$

$$PRD_j = SF_m - SF_j \qquad (9)$$

where i is the analysis year, j is the scenario number, m is the number of scenarios with the highest SF and the other variables were previously defined.

The effects of reducing the number of fatalities were estimated for the adopted assumptions and scenarios in the field of socioeconomic development of the country and the transport policy in relation to road safety. The results of these estimates are presented in Figure 11 and Table 8.

In general, the results of the analysis presented in Table 8 can be summarized as follows:

- in all scenarios, in the analysed period, a decrease in the number of fatalities can be expected,
- the implementation of road safety measures according to Scenarios S.1 or S.2 may bring very good effects,
- the bad and very bad effects can be expected when the tasks are implemented according to Scenarios S.3 and S.4.

**Table 8.** A summary of the expected effects of a particular scenario.

| Road Safety Indicators | Unit | Year/Period | Scenario | | | |
|---|---|---|---|---|---|---|
| | | | S1 | S2 | S3 | S4 |
| Forecasted number of fatalities | F | fatalities/year | 2020 | 1850 | 2250 | 2850 | 3000 |
| | | | 2030 | 650 | 1150 | 1650 | 2100 |
| | | | 2050 | 100 | 200 | 300 | 750 |
| Forecasted value of road fatality rate | RFR | fatalities/1 m. inhab./year | 2020 | 49 | 59 | 75 | 80 |
| | | | 2030 | 17 | 30 | 45 | 58 |
| | | | 2050 | 2 | 5 | 10 | 20 |
| Expected total number of road fatalities by 2050 | SF | thous. inhab/33 years | 2018–2050 | 24.1 | 36.5 | 50 | 64 |
| Estimated number of people who can be saved from death compared to the S4 scenario | PRD | thous. inhab/33 years | 2018–2050 | 40.0 | 27.5 | 14.0 | 0 |

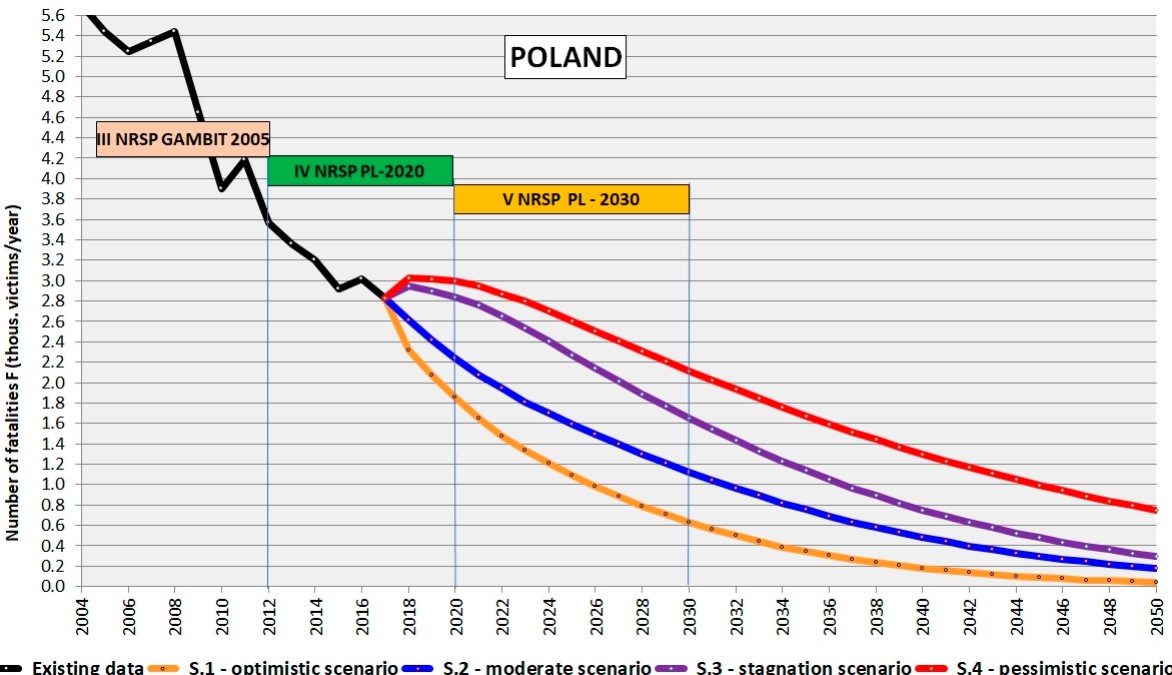

**Figure 11.** The forecasts of the number of fatalities in road accidents by 2050 in Poland for various road safety scenarios. Source: own study.

**S.1, an optimistic scenario**, presents the effect of applying an increased range of actions for road safety, which may result in maintaining a fairly high rate of change, i.e., about 170 deaths per year by 2030, and a reduction in the number of fatalities anticipated for 10 years of the programme implementation (2020–2030) may amount to 66%. Thus, the RFR indicator in 2030 may be close to the level of this indicator expected at this time in Sweden, the Netherlands and the United Kingdom. This scenario indicates the high possibility of achieving Vision Zero assumptions in 2050.

**S.2, a moderate scenario**, is a continuation but also a strengthening of many current road safety measures, which may result in maintaining a fairly high rate of change, i.e., about 130 fatalities by 2030, and the reduction in the number of deaths anticipated over the 10 years of programme implementation (2020–2030) may amount to 50%. This scenario indicates the possibility of achieving Vision Zero assumptions in 2050.

**S.3, a scenario of stagnation**, is a warning against a reduction in the scope of road safety measures, as the average rate of decreasing the number of fatalities will be around 45 deaths by 2030, and the reduction in the number of fatalities anticipated in the 10 years of programme implementation (2020–2030) may amount to 42%. However, this is too little to achieve the assumptions of the programme and to realize Vision Zero assumptions in 2050.

**S.4, a pessimistic scenario**, is a warning against abandoning or reducing the scope of road safety measures, as the average rate of decreasing the number of fatalities will be about 45 deaths per year by 2030, and reducing the number of fatalities anticipated within 10 years of programme implementation (2020–2030) can reach only 29%. This is far too little to achieve the programme's assumptions and to realize the Zero Vision assumptions in 2050.

Based on the history of current road safety measures, the implementation of activities according to the S.3 stagnation scenario seems the most likely. Unfortunately, this scenario will not ensure the achievement of the EU's strategic goal by 2030 (a reduction in the number of fatalities), so additional measures will be necessary to achieve these strategic goals, for example according to the S.2 moderate or S.1 optimistic scenarios.

Taking into account the assumptions of the 5th European Union Road Safety Programme for 2021–2030 as well as the analyses and forecasts of changes in the number of road fatalities in Poland, it is proposed to initially adopt the following main objective of V NRSP PL-2030: reducing the number of fatalities to approx. 1200.

The conducted analysis concerned only the impact of the analysed scenarios of road safety measures on the change in the number of fatalities. A similar analysis should be made by examining the impact of these activities on the change in the number of serious injuries.

*5.4. Directions of Activities Strengthening the Implementation of Vision Zero in Poland*

The main pillars of Vision Zero are the ethical behaviour of participants, facts and research and shared responsibility. The investigation into the Vision Zero intentions requires, as demonstrated at the Vision Zero conference in Stockholm in 2017 [62], the application of new ideas, new technologies and management systems including road user behaviour, innovative vehicles, safe road infrastructures, mobility management and the development of road safety system. The results of the research and analyses carried out in this work indicate five basic directions of strategic activities and offer many examples of operational activities for strengthening the possibility of implementing Vision Zero assumptions in Poland.

1. The activities aimed at the development of the road safety system: the adjustment of legal regulations to new challenges; the development and implementation of national, urban and regional level road safety programmes; and taking into account the activities of non-governmental organizations and social movements

2. The activities aimed at the change of road user behaviour: the use of an automatic key to block the driver under the influence of alcohol (alcolock), the development of automatic supervision and speed management (speed cameras, adaptive speed management systems such as Intelligent Speed Adaptation ISA, pedestrian and bicycle protection devices and new driver training systems)

3. The activities aimed at the development of innovative vehicles: the common use of winter tires; the development of devices supporting driver's actions (maintaining given speed and distance and detecting conflict situations); and the development and implementation of autonomous vehicles, electric and hybrid vehicles, car sharing and communication between vehicles and external devices (with other vehicles (V2V), road infrastructures (V2X) and traffic control systems (V2C)).

4.　The activities aimed at the development of modern and safe road infrastructure: the elimination of frontal collisions by separating the roadway (more common use of 2+1, 2×2 cross sections), the elimination of side collisions by the use of safe intersections (roundabouts and intersections with traffic lights), the use of new and safer types of interchanges, the use of road safety devices (barriers, crash cushions and fencing) and devices protecting vulnerable road users (sidewalks, cycle routes and pedestrian crossings), the development of autonomous and electric vehicle-friendly infrastructures, and taking the opportunities of Intelligent Systems Transport. In order to achieve this, it is necessary to

　　a.　improve the rules and guidelines for safe road design;
　　b.　develop new technologies and use building materials and construction elements with high durability and low maintenance requirements, guaranteeing high safety levels and efficiency (taking into account the object's life cycle); and
　　c.　develop new materials, technologies and construction elements that will enable a higher level of safety for road users.

5.　The activities aimed at the development of mobility management: traffic zoning, the propagation of shared spaces, the elimination of car traffic from central areas (charges, collective transport, bicycle transport and city bypass) and the application of new forms of urbanization (techno city and eco city).

The implementation of the activities indicated above should enable Poland to achieve Vision Zero assumptions; however, their selection and prioritization should be supported by an analysis carried out using the scenario method.

## 6. Discussion

The research and analyses helped to obtain complete or at least partial answers to the questions stated in Section 1.

**Approaches to road safety programming (question RQ1).** The results of the analyses (Sections 2, 3.1 and 4.2) prove that the system approach is more effective and comprehensive than the traditional approach.

**The general concept of changes in the level of road safety in the country (question RQ2).** The results of the self-conducted studies [8,9], confirmed by the results of other authors research [2–7,63], authorized the adoption of the concept of changes in the level of road safety in Poland as a generalization of the concept for any country. According to this concept, the level of road safety in the country depends on the level of socioeconomic development. Assuming that RFR is a normalized measure of mortality on roads in the analysed country, the level of road safety changes in a non-linear way depending on changes in the level of socioeconomic development (Figure 1).

In the range of very low and low levels of the country's socioeconomic development, together with the increase in the residents' income, their mobility increases; there is a growth in the motorization level and the density of paved roads, and this causes a rapid increase in the number of road fatalities from zero to the maximum.

On the border between very low and low levels of the country's socioeconomic development, there is a breakpoint in the RFR, which is caused by the decreasing rate of motorization growth, the beginning of the implementation of the road safety system (the development of the legislation system, education and appointing the leader), safety management methods (the development of the supervision system and road safety programmes) and a gradual change in the road users behaviour (reduced appetite for risk: lower speeds, the use of seat belts and no alcohol consumption before driving).

In the range of the medium level of the country's socioeconomic development, there is a rapid decline in the number of road fatalities. This is caused by factors including stabilization in the motorization growth, the paved roads' density and the mobility of residents; an increase in the density

of higher standards roads (express roads and motorways); the development of national and democratic institutions, thus reducing the level of corruption; the development of the health care system; and the development of the safety culture (using seat belts and reducing alcohol consumption).

In the range of high and very high levels of country's socioeconomic development, the road fatality rate tends asymptotically to zero. It is caused by the fact that the society is getting richer and the life of a road user is becoming more valuable [8].

The breakpoint in the curve describing the change in the number of fatalities is debatable. Some researchers [2–6,41,64,65] believe that the breakpoint is also noticeable in the case of RFR and occurs depending on the changes in the MR (in the range of 170–250 vehicles/1 thous. inhab.) or the GNIPC (national income growth) equal to 5.0–8.0 thous. USD (USA dollars). According to the authors' research, the breakpoint in the RFR occurs in the GDPPC range 10–18 thousand ID (International dollars), with a tendency to lower this value depending on the possibility of learning from more advanced countries (Figure 12). Developing countries benefit from the experience of the developed countries, invest more in the improvements of road safety, implement new solutions and regulations and thus reduce mortality in road transport.

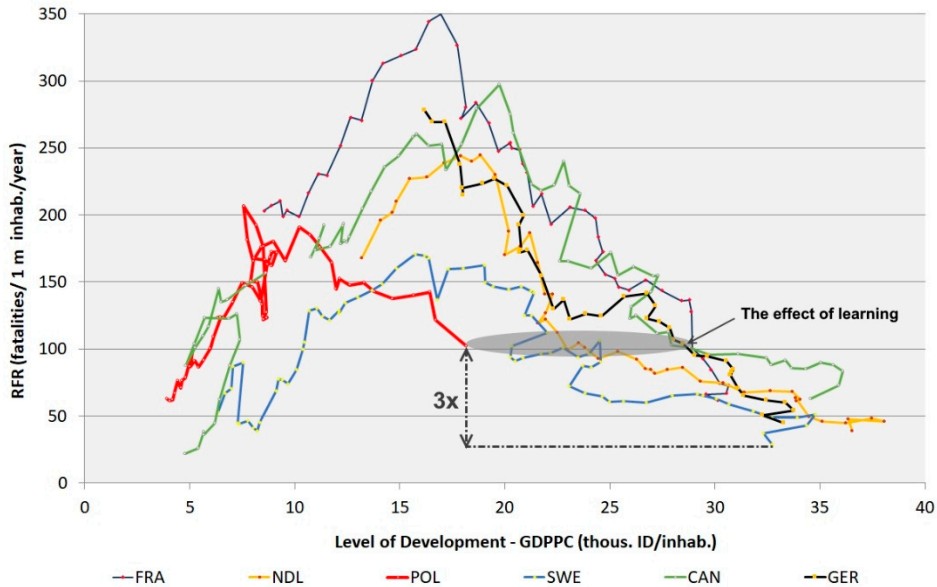

**Figure 12.** The model describing the influence of "the effect of learning" on the country's road safety level.

**Factors (variables and descriptors) included in the strategic forecasting of road safety measures (RQ3 question).** The analyses of factors affecting the changes of road safety indicators at the national level, carried out by the authors [8,9] and other researchers [3,4,6,7], show that the most important macro factors (as presented in Sections 3.3 and 5.2) are

- in the case of the transport system environment, the demography (population P), the economy (GDPPC) and a country's functional and organisational system (health system LEI, education system EDI and corruption level CPI), and
- in the case of road transport, the distance travelled by vehicles (VTK), the density of paved roads (DPR), the density of motorways and express roads (DME), the enforcement system quality (number of speed cameras FV) and the average road user behaviour (seatbelt usage USB and alcohol consumption ACPC).

**Methods used in the strategic analysis of the development of a country's road transport system, in the aspect of road safety (RQ4).** As shown in Sections 5.1 and 5.2, the scenario method used for strategic road safety programming should meet specific requirements. It should apply to long analysis periods, analyse the structural changes of key variables, ask questions and use theoretical

grounds and databases to determine the changes in key variables. The scenario method should be implemented according to the framework procedure in two stages: stage 1 scenario setting (including the identification of objectives and research areas, trend analyses, the analysis of opportunities and threats to achieve the objectives and the development of scenarios for strategic actions) and stage 2 scenario assessment (including analyses and prognostic simulations; evaluating the effectiveness of proposed groups of activities under different social, economic and technical conditions; and recommendations for the program).

**Strategic goals in selected reference periods (2020, 2030 and 2050) (PQ1).** The strategic analyses carried out for the purposes of road safety programming in future decades (described in Section 5.2) have shown that

- in all scenarios (except for the initial period for Scenarios S.3 and S.4) of the strategic actions, a decrease in the number of road fatalities is observed in the analysed period;
- in 2020, there is a risk of not achieving the objectives of the IV NRSP, as the predicted number of fatalities is in the range of 1850–3000 (compared to the target of 2000 fatalities);
- in 2030, there is a risk of not achieving the objectives of V NRSP, as the predicted number of fatalities is in the range of 650–2100 (compared to the initial target of 1000 fatalities). Thus, the goal of 1200 fatalities is the recommended target, taking into account a possible reduction that results from implementing scenario S2; and
- in 2050, a reduction in the number of fatalities to 100–750 can be expected. Taking into account the possibilities for implementation of strategic actions under Scenarios S.2 and S.3, this range narrows down to 200–300 fatalities. These numbers are close to the national expectations, but far from the recommendations of the European Commission.

**Recommended groups of strategic activities (PQ2).** The analyses (Section 5.3) helped to determine a set of additional operational activities to strengthen the strategic activities and activities implementing Vision Zero. The identified groups include the activities aimed at the development of the road safety system, the change of road user behaviours, the development of innovative vehicles, the development of modern and safe road infrastructures and the development of mobility management.

## 7. Conclusions

The following conclusions were made on the basis of the conducted research:

1. The country's socioeconomic development has a significant impact on the road safety level, and the most important factors include the gross domestic product, the mobility of residents, the level of the organizational system development (education, health care system and corruption level), the level of safe infrastructure development and changes in traffic behaviour (speed, seat belts and alcohol consumption).

2. The adopted model of road safety changes, and the proposed method of estimation of the number of fatalities constitute a substantive basis for formulating tasks and selecting priorities and directions of activities for the next road safety programming periods in Poland and in other countries.

3. The effectiveness of the undertaken actions towards a higher level of road safety in Poland is the result of many factors. Both the current state of scientific knowledge and the experience of countries with the highest level of road safety indicate that the optimal effects can be achieved by applying an ambitious vision and a systemic approach to achieving its objectives. The basic principles of such an approach indicate the key role of a precisely defined, clear philosophy of action based on scientific foundations, not myths and common opinions.

4. Despite the fact that many road safety programme activities are carried out, the implementation of the NRSP 2020 objectives is at risk as the results are so far less than expected: the pace of the decline in the number of fatalities is lower than expected and the pace of decline in the number

of seriously injured is very small, almost imperceptible. Therefore, undertaking actions by the European Commission in order to reduce the number of seriously injured, which is about ten times more than the fatalities, should be considered a proper and strongly expected solution.

5.   The application of the scenario method to assess the strategic objectives and directions of strategic activities helped to determine how the method should be used for road safety analyses. In addition, the method helped to clarify the objectives and to assess the effectiveness of the scenarios of strategic activities.

6.   The analysis shows that with the use of supportive and advanced measures, it is possible to significantly reduce the number of fatalities in subsequent programming periods and to achieve the Vision Zero assumptions in several dozen years.

7.   Poland's experience indicates that systematic political changes have a significant positive impact on the level of socioeconomic development, which also has a positive impact on road safety.

8.   In the case of Poland and other Central and Eastern Europe countries, accession to the EU brought significant benefits in terms of reduction in the number of fatalities. This was due to

a.   access to structural funds, which are aimed at supporting the social and economic development of regions of European Union countries and in particular the development of road infrastructure;

b.   the implementation of national requirements to fairly stringent European standards and in the field of road safety (Directive 2008/196/EC);

c.   access to international research programs and projects, research infrastructure and innovative technologies;

d.   the improvement of quality of life resulting from the implementation of EU's safety and environmental protection standards;

e.   access to the EU's procedures, schemas and patterns, which, if used efficiently, positively influence the progress in road safety improvement;

f.   the implementation of EU's transport development strategies and road safety programmes; and

g.   the EU's annual pressure on individual member states through publishing the rankings and analysing the progress in achieving road safety goals.

9.   Polish experiences in road safety programming were the basis for the preparation of the concept of the Integrated Transport Safety System [49].

**Author Contributions:** K.J. provided supervision and was the main editor of the paper. The methodology was developed by K.J and J.Ż. The editing of the article was done by A.R. The authors of particular sections were K.J. for Sections 1, 3.3, 5.1–5.3, 6 and 7; M.B. for Sections 4.1–4.3, 5.4 and 7; J.Ż. for Section 3.2; A.R. for Sections 1, 2, 4.3 and 7; W.K. for Sections 5.4 and 7; and J.O. for Sections 5.4 and 7.

**Funding:** This research received no external funding.

**Conflicts of Interest:** The authors declare no conflict of interest.

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
