# Peer review of "Experiences and Challenges in Fatality Reduction on Polish Roads"

_sustainability, doi:10.3390/su11040959_

Round 1
Reviewer 1 Report
The paper is interesting, well written and easy to understand. It provides an extended bibliography very useful for technicians and researchers. The parts related to the evolution of the road safety are very interesting and they contain an intelligent analysis of the condition of Poland. The part including the analysis of the different scenarios is interesting but, in my opinion, it is not completely clear. While, it is very well described the modality to consider the factors related to the economic condition of the country included in the formulas for estimating fatalities, it is not clear how the factors related to the implementing measures change in each scenario.
Finally, two minor remarks:
- many acronyms are not explained or they are explained after the sentence where they are written the first time (for example lines 21, 50, 140, 154). In my opinion, they have to be explained immediately when they appear for the first time.
- Line 121: what are the 4th, 5th, and 6th E?
Author Response
Dear Reviewer,
Thank you for your valuable review. We have made changes to our article to take account of your suggestions. Below please find the answers to your specific remarks:
“The paper is interesting, well written and easy to understand. It provides an extended bibliography very useful for technicians and researchers. The parts related to the evolution of the road safety are very interesting and they contain an intelligent analysis of the condition of Poland.
The part including the analysis of the different scenarios is interesting but, in my opinion, it is not completely clear. While, it is very well described the modality to consider the factors related to the economic condition of the country included in the formulas for estimating fatalities, it is not clear how the factors related to the implementing measures change in each scenario.”
Taking into account your remark, we added comment and table containing information on how particular factors change in each scenario. Both are included in Section 5.2.
“Finally, two minor remarks:
- many acronyms are not explained or they are explained after the sentence where they are written the first time (for example lines 21, 50, 140, 154). In my opinion, they have to be explained immediately when they appear for the first time.”
The text was corrected according to the remark – the acronyms were explained wherever the explanation was missing.
“- Line 121: what are the 4th, 5th, and 6th E?”
This was clarified in the revised manuscript. While 3 E’s is a road safety concept building on 3 road safety components: Engineering, Enforcement, Education, 4 E’s concept include additionally Emergency as the fourth component and finally, in the 6 E’s road safety concept, Encouragement and Economy are added as the 5th and 6th component.
Reviewer 2 Report
The paper is interesting and topical. It is clear that the authors are long-term experts, who push the national activities forward.
However, I have problem with one important part of the paper - the one related to the Figure 6. As very well phrased by prof. Ezra Hauer in the last sentence of his TZD white paper in 2010 (Lessons Learned from Other Countries): "It is tempting to attribute the decline in fatalities and injuries to the actions taken. Doing so may not be fully justifiable." In other words - attributing the variations in time series to "probable" reasons (as done in length on pages 13-16), may be not more than a "wishful thinking". Instead statistical methods should be applied, typically interrupted time series. For inspiration, see e.g.:
https://doi.org/10.1016/j.aap.2015.07.022
http://doi.org/10.1016/j.jsr.2014.07.001
https://doi.org/10.1080/17457300.2017.1341935
Minor issues:
lines 41-42 ...please explain the concept of "breakpoint"
You use terms "Poland’s road safety" or "Polish road safety". Please use one form consistently.
Term "fast traffic roads" in Fig. 6 is not typical - please change according to common terms.
Author Response
Dear Reviewer,
Thank you for your valuable review. We have made changes to our article to take account of your suggestions. Please find the answers to your specific remarks below:
“The paper is interesting and topical. It is clear that the authors are long-term experts, who push the national activities forward.
However, I have problem with one important part of the paper - the one related to the Figure 6. As very well phrased by prof. Ezra Hauer in the last sentence of his TZD white paper in 2010 (Lessons Learned from Other Countries): "It is tempting to attribute the decline in fatalities and injuries to the actions taken. Doing so may not be fully justifiable." In other words - attributing the variations in time series to "probable" reasons (as done in length on pages 13-16), may be not more than a "wishful thinking". Instead statistical methods should be applied, typically interrupted time series. For inspiration, see e.g.:
http://dx.doi.org/10.1016/j.aap.2015.07.022
http://dx.doi.org/10.1016/j.jsr.2014.07.001
http://dx.doi.org/10.1080/17457300.2017.1341935”
Thank you for invoking the topic, forecasting road safety indicators for long time horizons is indeed a significant problem. So far, not much work can be found regarding the use of strategic analysis for road safety programming. The recalled approach of E. Hauer and the other researchers can be applied to short-term forecasting. However, in long-term forecasting it is ineffective. Therefore, as road traffic engineers, using the applications in transport planning, we have attempted to use a scenario approach to assess the use of a group of strategic activities in the road safety programming for long time horizons. Our approach, supported by the results of literature studies and case studies for Poland, is presented in the extended section 5 (in particular, sections 5.1 and 5.2), therefore, we do not repeat the same arguments in this explanation. By expanding section 5 with a new issue, application of scenario methods, we would like to start a discussion on the application of the methods to strategic analyses supporting road safety programming.
“Minor issues:
lines 41-42 ...please explain the concept of "breakpoint"”
The breakpoint concept was better explained in the manuscript.
It was proved in many studies cited in the manuscript, that the road fatality rate (RFR) changes nonlinearly with the level of socio-economic development of a country, measured with Gross Domestic Product per capita (GDPPC). In the initial phase of socio-economic development the RFR increases, until reaching the breakpoint, which occurs at the low level of GDPPC and when the RFR is the highest. When the breakpoint is exceeded, RFR decreases with further GDPPC increase. This relation between road safety level and socio-economic development is better described now in the revised manuscript – Section 1, and presented in Figure 1.
You use terms "Poland’s road safety" or "Polish road safety". Please use one form consistently.
The suggestion was taken into account and the term was unified (“Poland’s road safety term” is used consistently in the manuscript).
Term "fast traffic roads" in Fig. 6 is not typical - please change according to common terms.”
The term “fast traffic roads” was changed with “motorways and express roads”.
Round 2
Reviewer 2 Report
Thank you for the revision. I can see you addressed the comments. Especially thanks for extending the section 5 in order to better explain the applied concepts. Now I see the paper as ready for publication.